## Parent material geochemistry - and not plant biomass - as the key factor shaping soil organic carbon stocks in European alpine grasslands

Annina Maier<sup>1,\*</sup>, Maria, E. Macfarlane<sup>1, 2,\*</sup>, Marco Griepentrog<sup>1</sup>, and Sebastian Doetterl<sup>1</sup>

<sup>1</sup>Department of Environmental Systems Science, Swiss Federal Institute of Technology (ETH) Zurich, 8092 Zurich, Switzerland

<sup>2</sup>Department of Civil and Environmental Engineering, Massachusetts Institute of Technology (MIT), 02139-4307 Cambridge, MA, USA

\*These authors contributed equally to this work.

Correspondence: Annina Maier (annina.maier@usys.ethz.ch)

Abstract. Soils represent the largest terrestrial carbon (C) reservoir on Earth. Within terrestrial ecosystems, soil geochemistry can be a strong driver of plant-soil-carbon dynamics, especially in young, less weathered soils. Here, we investigate the impact of potential plant biomass input, soil fertility parameters, and soil organic carbon (SOC) stabilization mechanisms on the distribution of SOC in European alpine grasslands across gradients of geochemically distinct parent materials. We demonstrate that SOC stock accrual and persistence in geochemically young soils, with fraction modern ( $F^{14}C$ ) values ranging from 0.77–1.06, is heavily dependent on soil mineralogy as a result of parent material weathering, but is not strongly linked to plant biomass. We show potential differences in the importance of geochemical variables and SOC stabilization mechanisms, with the microaggregate soil fraction contributing  $\geq 50$  % to bulk SOC in the majority of cases. We further show that concentrations of Fe, Al and Mn pedogenic oxides coincide with SOC stock magnitude across an alpine soil geochemical gradient, where SOC stocks range between 8.1–23.2 kg C m<sup>-2</sup>. Our results highlight that soil fertility and soil mineralogical characteristics, which govern plant C inputs and control C stabilization respectively, play equally crucial roles in predicting SOC contents in alpine soils at an early development stage.

Copyright statement. This work is distributed under the Creative Commons Attribution 4.0 License.

#### 1 Introduction

Mountain ecosystems are experiencing intense climate warming with temperatures increasing by 0.4–0.6 °C per decade (Pepin et al., 2015; Vitasse et al., 2021). This warming may result in the extension of vegetation zones and thermal limits of plant life by 300–600 m in elevation (Hagedorn et al., 2019). Indeed, an increase in alpine plant productivity has already been detected in mountains of central Asia and the European Alps (Choler et al., 2021; Anderson et al., 2020; Carlson et al., 2017). Approximately 77 % of the area of European Alps has experienced greening above the treeline within the last four decades alone.

However, while many studies have focused on the responses of aboveground mountain vegetation to climate warming (Rumpf et al., 2022; Nomoto and Alexander, 2021; Steinbauer et al., 2018), less attention has been given to belowground changes and the impact of ongoing soil weathering, despite more than 90 % of alpine ecosystem carbon (C) being stored in soils above the treeline (Körner, 2003a). In this regard, climate's interaction with parent material and geochemistry are important factors governing rates of soil weathering and remain understudied controls on C cycling and stabilization in alpine soils (Jenny, 1994; Barré et al., 2017; Yang et al., 2020).

Plants residues represent the main source of organic carbon (OC) entering the soil system. The extent of this input depends on variations in above- and belowground biomass production, which are influenced by climate zone, elevation and soil parent material (Körner, 2003a; Hagedorn et al., 2019; O'Sullivan et al., 2020; Augusto et al., 2017; Paoli et al., 2008). Through weathering processes, parent material delivers nutrients into the soil solution, which are a limiting factor for plant growth, especially in young alpine ecosystems (Möhl et al., 2019; Körner, 2003a). Weathering also leads to increased formation of reactive secondary minerals, such as expandable clays and short-range order minerals (SRO)/oxy(hydr-)oxides. Together with cations they contribute to SOC stabilization and persistence through direct interaction and complexation of organic molecules, a mechanism which is highly influenced by parent material geochemistry and soil weathering (Rowley et al., 2018; Solly et al., 2020). SOC persistence is further promoted by aggregate formation and stability, which are enhanced by clay content, Ca<sup>2+</sup> ions (in high pH soils), and pedogenic oxy(hydr-)oxides (Olagoke et al., 2022; Rowley et al., 2021; Kirsten et al., 2021). Recent studies highlight pedogenic Fe and Al oxy(hydr-)oxides as key drivers of SOC stabilization, often surpassing silicate clay minerals in importance (Fang et al., 2021; Doetterl et al., 2015; Mao et al., 2020; Yang et al., 2020; Kirsten et al., 2021; Reichenbach et al., 2021). Their poorly crystalline forms enhance aggregation (Baldock and Skjemstad, 2000; Duiker et al., 2003; Kaiser et al., 2016) and provide reactive surfaces for SOC stabilization via ligand exchange (Lützow et al., 2006). Many of the above-described SOC stabilization mechanisms are assumed to be underdeveloped in alpine soils that are geochemically younger, less altered, and have not undergone intense weathering in the past. This is linked to the climate conditions found in alpine ecosystems which typically lead to a slower - or limited - formation of alpine soils compared to lowland soils (Körner, 2003b; Doetterl et al., 2018). However, few studies have examined the relative importance of plant biomass-derived C inputs into the soil compared to SOC stabilization potentials of alpine soils for SOC stock accrual.

The lack of interactions that enable SOC stabilization in alpine soils has been shown to result in a heightened vulnerability of OC stocks to changes in climate (Parker et al., 2015; Hagedorn et al., 2019; Walker et al., 2022; Garcia-Franco et al., 2024). However, climate warming will also induce an acceleration in soil weathering rates and improve climatic growth conditions of high-altitude vegetation, which may also lead to the development of larger SOC stocks (Rumpf et al., 2022; Muñoz et al., 2023). The rate and extent to which climatic change may influence the soil matrix and long-term SOC stabilization is strongly linked to the reactivity of the soil's mineral phase, which is largely governed by parent material (Doetterl et al., 2015). The majority of past work on the future of the alpine soil C cycle has focused on studying elevational SOC gradients studying vegetation patterns (Hitz et al., 2001; Guidi et al., 2024) as well as soil parameters in relation to different alpine habitats

(Canedoli et al., 2020; Cao et al., 2013). A systematic analysis of the interaction between the pedosphere and biosphere in shaping regional-scale alpine SOC stocks across geochemically contrasting parent materials remains missing. Exploring variations in soil weathering, biomass inputs, and SOC stocks across different parent materials will likely improve our understanding of future C cycling in alpine environments. Accordingly, we examine soils formed on five geochemically distinct parent materials (Dolomite, Flysch, Gneiss, Greenschist, Marl) in the European Alps to assess how differences in parent material, and consequently soil geochemistry, influence (i) the magnitude and variability of potential biomass inputs (above- and belowground) and (ii) SOC stocks. To do so, we pair the quantification of SOC stocks under comparable climatic and environmental condition with an assessment of the persistence of soil C, via radiocarbon (<sup>14</sup>C) analyses and how different soil stabilization mechanisms contribute to bulk SOC, via physical soil fractionation.

First, we hypothesize that i) soils developed on nutrient-rich parent materials will exhibit the highest biomass inputs and potentially also the highest overall SOC stocks, if soils can stabilize these inputs (e.g., through high clay content or pedogenic oxides). More specifically, we expect that ii) soils with a higher amount of pedogenic oxides—and not the amount of clay as a particle size—will lead to greater SOC stocks, because of the enhanced stabilization potential of those minerals (Rasmussen et al., 2018). Lastly, we expect iii) soils that are more strongly weathered will not only show higher SOC stocks but also older <sup>14</sup>C values, hinting at higher persistence, by having developed a wider portfolio of efficient mineral-related stabilization mechanisms, resulting in better protection of SOC against microbial decomposition (Kleber et al., 2015; Torn et al., 2013).

#### 2 Materials and methods

#### 2.1 Study area



The study area was located within grasslands and meadows of the European Alps, along a geochemical gradient of soil parent materials. Three out of five suitable study sites with Flysch, Gneiss, and Greenschist as parent materials were located in the Eastern part of Switzerland (i.e., canton of Graubünden). The two remaining study sites with Dolomite and Marl as parent materials were sampled in Germany and Austria, respectively (Fig. 1). The specific sampling sites were chosen based on geological maps (Federal Office of Topography swisstopo, 2020; Federal Institute of Geosciences and Natural Resources, 2020) (Fig. 1). Sampling sites varied primarily in parent material while elevation, aspect, and climatic factors were kept as constant as possible. The elevation of all sampling sites is within a range of 2000–2300 m a.s.l. There was little variation in climate among the sites. The mean annual temperature (MAT) of the sites ranges between 1.4–2.8 °C (Hagedorn et al., 2010; Bassin et al., 2013) and the mean annual precipitation (MAP) varied from a minimum of 1050 mm in Davos (CH) to a maximum of 1300 mm in the Zugspitze region (DE) and in Allgäu (AT) (VAO (Virtual Alpine Observatory), 2020). A more detailed description of the sampling sites' locations as well as their geochemical and climatic characteristics are summarized in Table 1.

All examined parent materials are distinct in their geochemical composition and formed clearly distinguishable soils (Table 1, Fig. 2). To classify each sampling site's soil type, soil profiles were dug in the center of the sampling plots and described

**Figure 1.** Locations of the soil sampling sites developed on distinct parent materials. This map was produced using Copernicus WorldDEM-30 © DLR e.V. 2010-2014 and © Airbus Defence and Space GmbH 2014-2018 provided under COPERNICUS by the European Union and ESA; all rights reserved (European Space Agency (ESA), 2019).

according to the Manual of Soil Mapping (Eckelmann et al., 2006). Soil classification was done following the World Reference Base for Soil Resources (WRB, 2015). The soil developed on Dolomite was classified as a Dolomitic Leptosol. This soil did not possess any mineral horizons but had an average  $12.9 \pm 2.2$  cm thick Oh horizon, which was lacking in all other sites' profiles. The Dolomite site's Oh horizon lies directly on top of the bedrock (R) horizon, which consists of calcareous material. All other sites developed Ah and Bw horizons of varying thicknesses on top of a Cw horizon. The soil developed on Flysch reached a depth of  $37.2 \pm 0.2$  cm and was classified as an Argic Umbrisol. The soil on Gneiss developed  $53.7 \pm 1.8$  cm as a Spodic Umbrisol, due to the accumulation of Fe and Al oxides in subsurface horizons. The soil on Greenschist developed into  $39.9 \pm 3.2$  cm depth as an Umbrisol. The soil on Marl developed into  $67.2 \pm 1.2$  cm depth as a Cambisol, (calcaric), containing some calcaric material in its Cw horizon. The rooting depths of all sites reached until at least the end of their Bw horizons, except the Gneiss site which reached approximately half of the Bw horizon depth. Further information on the examined soils' biogeochemical and physical characteristics can be found in Table 4.

#### 2.2 Sampling design




Soil and biomass sampling was carried out during the vegetation growth period from late June to early September, in 2020. At each site, four field replicate plots of 25 x 25 m were chosen. The plots were considered suitable when their slopes were between 40–60 %, their aspects were south-facing (with slight variation across sampling sites from SE to SW), and no depressions were found. Ten soil monoliths were sampled per field replicate plot in a depth-explicit way, respecting horizon boundaries and with the aim of reaching the least weathered horizon (C horizon) at each sampling site. In that sense, the sampled soil horizons may not align with the same soil depths, across sites. Horizon-explicit sampling was conducted, as the main interest lay in capturing the functional and developmental variability of the different horizons as the result of longer-term pedogenetic processes across

Table 1. Overview on parent materials, geochemical characteristics, locations, elevation ranges, MAT and MAP of the study sites.

| Parent material | Geochemical characteristics              | Location                               | Elevation (m.s.l.) | MAT (°C) | MAP (mm) | Slope (%) | Aspect (-) |
|-----------------|------------------------------------------|----------------------------------------|--------------------|----------|----------|-----------|------------|
| Dolomite        | Carbonatic                               | Wettersteinkalk<br>mountain range (DE) | 2000–2300          | 1.4      | 1300     | 44 ± 2    | S          |
| Marl            | Carbonatic, sedimentary                  | Allgäu Alps (AT)                       | 2180–2230          | 1.4      | 1300     | 50 ± 6    | SW         |
| Gneiss          | Siliceous, intermediate metamorphic      | Dischma valley,<br>Davos (CH)          | 2080–2200          | 1.4      | 1050     | 71 ± 2    | S          |
| Greenschist     | Siliceous, mafic<br>metamorphic, Mg-rich | Crap da Radons,<br>Bivio (CH)          | 2160–2200          | 2.8      | 1200     | 60 ± 4    | SE         |
| Flysch          | Siliceous, intermediate sedimentary      | Curtegns valley,<br>Savognin (CH)      | 2180–2220          | 2.8      | 1200     | 48 ± 2    | SE         |

Figure 2. Average  $\pm$  standard errors of total depth of whole soil profiles, divided into individual horizons (cm), average rooting depth (cm) and presence of carbonates (No/Yes) are shown  $\pm$  standard errors, across all sites. Soil profiles are displayed along a relative gradient in Fe<sub>DCB</sub>/Fe<sub>tot</sub>. Precise values for these indices per soil horizon can be found in Table 4. A description and the calculation of this index can be found in Sect. 2.4.1. Note that the Dolomite site's Oh horizon is an organic horizon, however it is displayed in a similar fashion to all other mineral horizons for illustrative purposes. Same respective lowercase letters denote total soil profile development depths, that are not statistically significant from one another.

the sites. Each horizon of the soil monoliths was sampled using a Kopecky cylinder. Then, the ten soil samples sampled by Kopecky cylinders per horizon and plot were merged into one composite sample per field replicate plot. Additionally, an Edelman auger was used to obtain samples of the weathered bedrock (Cw) and of the parent material, i.e., unweathered bedrock

(R). Further, at each field replicate plot five smaller plots were chosen as representative field replicates for aboveground plant biomass sampling. Each of these replicates covered an area of 0.2 x 0.2 m for the Marl and Flysch sites and 0.4 x 0.4 m for the Gneiss, Greenschist and Dolomite sites. From these smaller plots, the entire aboveground biomass was sampled. Ultimately across all five study sites a total of 100 observations were sampled, consisting of 46 soil, 18 parent material, 18 above- and 18 belowground plant biomass samples.

#### 115 2.3 Laboratory analyses

#### 2.3.1 Sample processing




All collected soil and aboveground biomass samples were oven-dried at 40 °C for 96 h and their dry weights noted. Total aboveground plant biomass was then split into woody *vs.* non-woody biomass and their respective weights noted. Dried soil samples were split and sieved through a 2 mm sieve. Large roots that did not pass through the 2 mm mesh were collected and weighed. Fine roots that passed the 2 mm mesh were collected with tweezers. The fine soil fraction (< 2 mm) gained from sieving was used for further analyses. Moreover, an aliquot of each fine soil sample was milled (Mixer Mill MM 200, Retsch GmbH Haan, Germany) as a requirement for certain analyses such as carbon (C) and nitrogen (N) contents, elemental composition, bioavailable phosphorus, cation exchange capacity, sequential pedogenic oxide extraction and <sup>14</sup>C analyses. Further, a representative aliquot of each site's aboveground plant biomass was homogenized in two steps for C and N analyses; the plant material was coarsely shredded in a blender and then milled. In addition, a representative aliquot of each site's belowground biomass (coarse and fine roots) was milled (Mixer Mill MM 200, Retsch GmbH Haan, Germany).

#### 2.3.2 Physical soil fractionation

Physical fractionation of bulk soil samples was conducted on three of the four field replicates per sampling site, with the exception of the Marl site. For this site only two of the four field replicate plots were included in the physical fractionation scheme. The respective excluded sampling plots displayed a parent material geochemistry that did not correspond with the geochemical composition of a typical Marl, exhibiting markedly higher Si contents. The fractionation corresponded to a simplified version of Six et al. (2000), as applied in Doetterl et al. (2018) using a microaggregate isolator installed on a sample shaker. First, a 20 g aliquot of bulk soil samples (< 2 mm) was weighed and submerged in water for 48 h to break down non-water-stable aggregates. Second, the samples were wet-sieved through a 250  $\mu$ m sieve, with the material > 250  $\mu$ m building the coarse particulate organic matter (POM) fraction. Macroaggregates were broken up using the microaggregate isolator and shaker. Subsequently, samples were sieved through a 53  $\mu$ m sieve to obtain the silt and clay (s+c) fraction (< 53  $\mu$ m) and the stable microaggregate fraction (MA) (53–250  $\mu$ m). Finally, the obtained fractions were analyzed for their N and OC content (Sect. 2.3.4). The procedural and conceptual interpretations of the obtained fractions can be found in Table 2. An overview on the fractionation results can be found in Table A1.

**Table 2.** Fraction names and abbreviations, procedural definitions and conceptual interpretations of SOC associated with the respective soil fractions. This table was adapted from Doetterl et al. (2018) and Doetterl et al. (2015).

| Fraction                                | Procedural definition | Conceptual SOC interpretation                                                                                            |
|-----------------------------------------|-----------------------|--------------------------------------------------------------------------------------------------------------------------|
| Coarse particulate organic matter (POM) | > 250 μm              | Unprotected SOC, non mineral-associated                                                                                  |
| Stable microaggregates (MA)             | 53–250 μm             | Microaggregate-associated SOC: physical separation of unprotected SOC through aggregation and organo-mineral association |
| Non-aggregated silt and clay (s+c)      | < 53 μm               | Non-aggregated silt- and clay-associated SOC: organo-mineral association of SOC and association with smallest aggregates |

#### 140 2.3.3 Determination of soil texture, pH, elemental composition and effective cation exchange capacity


Soil texture was determined according to Miller and Schaetzl (2012) using a laser diffraction particle size analyzer (LS 13 320 Laser Diffraction Particle Size Analyzer, Beckman Coulter, Brea, CA, United States). Following a modified protocol from Rayment and Lyons (2011), pH measurements were conducted by adding 15 ml of 0.01 M CaCl<sub>2</sub> to 3 g of soil sample. The samples were shaken horizontally for 10 minutes and measured after 24 h with a pH meter (Metrohm 713). X-ray fluorescence (XRF) spectrometry was applied to determine the composition of the heavier elements Si, Ti and Zr (Spectro XEPOS, Spectro Analytical Instruments, Kleve, Germany) in milled fine soil and parent material samples. The elements Na, K, Ca, Mg, Fe, Al and Mn were determined by inductively coupled plasma optical emission spectrometry (ICP-OES) (5100 ICP-OES, Agilent Technologies, Santa Clara, CA, United States). Here for, 1 g of milled fine soil samples were weighed into plastic tubes and were extracted with "aqua regia", i.e. 2 ml nanopure H<sub>2</sub>O, 6 ml hydrochloric acid (HCl, 37 %) and 2 ml nitric acid (HNO<sub>3</sub>, 65 %) for 150 min at 120 °C. The extracts were then diluted with nanopure H<sub>2</sub>O to a defined volume of 50 ml and filtered through 20 µm paper filters. Effective cation exchange capacity (CECeff) and the amount of exchangeable cations (Ca, Mg, K, Na, Al, Fe, Mn) were determined according to Hendershot and Duquette (1986) using BaCl<sub>2</sub>. In brief, 20 ml of 0.1 M BaCl<sub>2</sub> were added to 3 g of soil in a 50 ml centrifuge tube. These samples were then placed on a horizontal shaker for 2 h at a setting of 150 rpm, centrifuged at 2500 rpm for 10 min and filtered (42 µm) into 50 ml tubes. Extracts were then stored at room temperature before measurement. Shortly before measurement on the Inductively Coupled Plasma Optical Emission Spectroscopy (ICP-OES) (5100 ICP-OES, Agilent Technologies, Santa Clara, CA, United States), samples were diluted 1:10 with MQ and 0.1 ml of internal standard (in HNO3, 65 %). Sampling site specific patterns in soil texture, pH and CEC<sub>eff</sub> can be found in Table 4.

#### 2.3.4 Determination of organic carbon, nitrogen, bioavailable phosphorous content

The aliquouts from the milled, dried fine soil and plant samples mentioned above (Sect. 2.3.1) were analyzed for their OC and N content using a CN analyzer (Vario MAX cube, Elementar, Langenselbold, Germany). Soils with a pH > 6 were treated before the analysis to remove the carbonates. The carbonate removal was conducted according to Ramnarine et al. (2011) using

hydrochloric acid fumigation and adopted similarly as Peixoto et al. (2020). A beaker of 50 ml of 37 % HCl was prepared and placed in a desiccator. The soil samples were added for a duration of 72 h. Afterwards the beaker filled with HCl was replaced with NaOH pellets for the removal of residual HCl vapor for additional 72 h. Each sample was measured without HCl treatment for accurate nitrogen contents Walthert et al. (2010).

The dry weights of the collected plant and root biomass samples, together with their OC and N contents were used to receive above- and belowground OC stocks and use them as proxies for potential C inputs into the soil (Sect. 2.4.2, Eq. 1). Regarding the choice to mix fine and coarse roots into joined, composite samples, we would like to acknowledge the differences in C:N ratios, decomposition mechanisms and functions that exist between coarse and fine roots (Zhang and Wang, 2015). However, fine root samples constituted only very small fractions of overall belowground plant biomass and our primary objective was to obtain an approximate estimate of respective belowground C and N contents. Further, we acknowledge that this approach does not allow for the separation of living and dead roots at the time of sampling or the assessment of belowground biomass productivity. Lastly, we recognize that the analysis of standing aboveground biomass stocks does not truly represent aboveground OC inputs into the soil, compared to e.g. quantification of effective litter inputs into the soil. However, it should allow for a snapshot of potential below- vs. maximum aboveground biomass inputs at the time of sampling, which is the main purpose of the data.

For the determination of bioavailable phosphorus (bio-P) content, soil samples were extracted with anion exchange membranes (VWR International, material no. 551642 S) in H<sub>2</sub>O for 16 h with a 4 cm<sup>2</sup> membrane per g of fresh soil. Prior to extraction, the membranes were activated with 0.5 M NaHCO<sub>3</sub>. Post extraction, the P retained on the membranes was extracted off the membranes during 1.5 h on a horizontal shaker by using 0.25 M H<sub>2</sub>SO<sub>4</sub>. Finally, the P contained in the resulting extracts was quantified with ICP-OES (5100 ICP-OES, Agilent Technologies). Parent material specific patterns in bio-P can be found in Table 4.

#### 2.3.5 Sequential extraction of pedogenic oxides





To identify reactive metal phases that can interact (and potentially stabilize) soil C, Fe, Al and Mn oxides were extracted sequentially using sodium pyrophosphate (PP) (Bascomb, 1968), ammonium oxalate (AO) (Dahlgren, 1994), and dithionite-citrate-bicarbonate (DCB) solutions (Mehra and Jackson, 1958). The extraction procedure is based on duplicate extractions, hence, each sample was extracted twice in the form of two subsamples which were merged for final measurement with ICP-OES (5100 ICP-OES, Agilent Technologies, Santa Clara, CA, United States). In brief, 0.5 g of 30 °C dried and milled bulk soil was extracted with a sodium-pyrophosphate (PP) solution at pH 10 (consisting of 0.1 M Na<sub>4</sub>P<sub>2</sub>O<sub>7</sub> x 10 H<sub>2</sub>O and 0.5 M Na<sub>2</sub>SO<sub>4</sub>) at a soil to solution ratio of 1:40 for 16 h on a horizontal shaker. Then the vials were centrifuged (Sigma 3–16 KL, 20 °C, 1700 × g) for 10 min. The supernatant was decanted, filtered (Whatman 41) and diluted to 50 ml with MilliQ. These extracts were stored at 4 °C prior to measurement on ICP-OES. The remaining soil residue was extracted with a 0.2 M ammonium oxalate (AO) solution (consisting of 0.2 M ammonium oxalate and 0.2 M oxalic acid dihydrate in a ratio of

1:31:1) at pH 3 and a soil to solution ratio of 1:40 for 2 h on a horizontal shaker in the dark to prevent photodegradation of the extract. Then the vials were centrifuged (Sigma 3–16 KL, 20 °C, 1700 × g) for 10 min. The supernatant was decanted, filtered (Whatman 41) and diluted to 50 ml with MilliQ. These extracts were stored at 4 °C prior to measurement on ICP-OES. The remaining soil residue was placed in a pre-heated water bath (70–75 °C) with 0.3 M sodium citrate and 1 M NaHCO<sub>3</sub>, at a soil to solution ratio of 1:22.5. To this 0.25 g sodium dithionite was added stepwise every 15 min until the solution was grey-coloured. After, 5–10 min of cooling outside the bath Then the vials were centrifuged (Sigma 3–16 KL, 20 °C, 4000 rpm) for 10 min. The supernatant was decanted, filtered (Whatman 41) and transferred into a clean vial. 0.5 ml of MilliQ were added to the leftover soil pellets and the vials were centrifuged again (Sigma 3–16 KL, 20 °C, 3000 rpm) for 10 min. The supernatants were transferred analogously into the vial containing the previous supernatant. This procedure was repeated once more. Then, 2.5 ml of magnesium sulfate was added to the soil pellets. Samples were vortexed, ultrasonicated and centrifuged again (Sigma 3–16 KL, 20 °C, 3000 rpm) for 10 min. The supernatants were transferred analogously into the vial containing the previous supernatants. These extracts were stored at 4 °C prior to measurement on ICP-OES.

We acknowledge that a release of elements from non-target, more crystalline minerals such as magnetite, maghemite, biotite, chlorite, muscovite and illite may occur to varying extent during any of the three sequential extraction steps. To which
degree is often unknown prior to extraction (Rennert, 2018). Therefore, the interpretation of pyrophosphate extracts as the
organo-metal complexed oxides is treated with caution. The term "pedogenic" oxides for Fe-, Al- and Mn- phases extracted in
the context of this extraction procedure, is applied for purposes of simplicity. For clarification, Table 3 provides an overview on
how the extractable fractions were interpreted. Site-specific patterns in the sum of Fe, Al and Mn pedogenic oxides, extracted
with PP and AO, can be found in Table 4.

**Table 3.** Abbreviations, definitions and interpretation of extracted Fe, Al, Mn pedogenic oxides following Carter and Gregorich (2008).

| Abbreviation                                              | Definition                              | Interpretation                                                                                                               |
|-----------------------------------------------------------|-----------------------------------------|------------------------------------------------------------------------------------------------------------------------------|
| Fepp, Alpp, Mnpp                                          | Pyrophosphate-extractable Fe, Al, Mn    | Primarily dissolved organo-metal-complexed Fe, Al, Mn oxides but may also include some ( $<$ 10 %) poorly crystalline oxides |
| Fe <sub>AO</sub> , Al <sub>AO</sub> , Mn <sub>AO</sub>    | Ammonium-oxalate-extractable Fe, Al, Mn | Poorly crystalline Fe, Al, Mn oxides and short-range order minerals                                                          |
| Fe <sub>DCB</sub> , Al <sub>DCB</sub> , Mn <sub>DCB</sub> | DCB-extractable Fe, Al Mn               | More crystalline Fe, Al, Mn oxides                                                                                           |

#### 2.3.6 Determination of $F^{14}C$




Composite samples of bulk soil for <sup>14</sup>C analyses were created by pooling similar masses per horizon of each sites' respective field replicates. From these composites, small subsamples between 2–32 mg of milled material, corresponding to *ca.* 300 µg C, were transferred into 0.025 ml tin capsules. Prior to measurement with the EA-AMS, samples were acidified to remove carbonates (Komada et al., 2008). <sup>14</sup>C analyses were conducted using an elemental analyzer coupled to a gas-ion-source

equipped accelerator mass spectrometer (EA-AMS) at the Laboratory of Ion Beam Physics (LIP) at ETH Zurich (Haghipour et al., 2019; Welte et al., 2018). Results of the  $^{14}$ C analysis are reported as fraction modern ( $F^{14}$ C) according to Reimer et al. (2004).  $F^{14}$ C values were compared across sites and soil horizons. The values were interpreted as a proxy of SOC persistence and as an indicator of longer term SOC stabilization and turnover dynamics. Hereby, lower  $F^{14}$ C values were interpreted as being more persistent, i.e. more stable, than higher  $F^{14}$ C values (Mathieu et al., 2015).  $F^{14}$ C values > 1 are defined as modern. We acknowledge that the measured bulk  $^{14}$ C values represent a mixture of different C pools with different stabilities/persistence and that these values we report may under- or overestimate soil C persistence (Torn et al., 2013; Trumbore, 2000).

#### 2.4 Calculations






#### 2.4.1 Calculation of weathering and fertility indices

Weathering and fertility indices including the total reserve of base cations (TRB) ( $\Sigma$ Ca, K, Mg and Na [g kg $^{-1}$ ]) and the ratio of DCB-extractable Fe to total Fe (Fe<sub>DCB</sub>/Fe<sub>tot</sub> [-]) were calculated for each horizon of every sampling site. TRB decreases with ongoing soil development, as these soluble cations are increasingly leached from the soil (Brady and Weil, 2008). The Fe<sub>DCB</sub>/Fe<sub>tot</sub> ratio relates total pedogenetially-transformed Fe phases extracted with DCB, to total Fe. This value increases with ongoing soil development as more Fe becomes pedogenetially-transformed to more crystalline oxide forms. These weathering/fertility indices were calculated based on measured values of ICP-OES measurements for TRB and on sequential pedogenic oxide concentrations and ICP-OES measurements for Fe<sub>DCB</sub>/Fe<sub>tot</sub>. Site-specific patterns in Fe<sub>DCB</sub>/Fe<sub>tot</sub> ratios and TRB can be found in Table 4. Note that while Dolomite is displayed as the site with the highest value for the weathering index Fe<sub>DCB</sub>/Fe<sub>tot</sub> (Fig. 2, Table 4), this value does not represent the true value for the in situ Dolomite soil. A past report has shown that most horizons of examined soil profiles in the Wettersteinkalk/Zugspitze region show signs of aeolian influence (Küfmann, 2008). Thus the Fe<sub>DCB</sub>/Fe<sub>tot</sub> value found at the Dolomite site rather reflects in situ weathering of deposited mineral dust inputs.

#### 2.4.2 Calculation of plant biomass and soil organic carbon stocks

Plant biomass OC stocks for above- and belowground biomass were calculated based on the plant biomass dry weight (DW) and the OC concentration values for above- and belowground biomass per sampling site. Plant biomass OC stocks will be interpreted as a proxy for potential plant biomass C input, as plant input rates were not measured directly within this study. Where plant biomass OC stock (BM  $OC_{stock_i}$ ) is defined as the plant biomass OC stock for above- or belowground (i) plant biomass following Eq. 1.

$$BM OC_{stock_i} (kg m^{-2}) = BM OC_{conc_i} (\%) \cdot BM DW_i (kg C m^{-2})$$

$$\tag{1}$$

The most commonly used method to calculate SOC stocks is based on bulk density, soil horizon thickness, and the SOC concentration values (Poeplau et al., 2017). However, this simple calculation often overestimates SOC stocks because it neglects the rock fragment fraction. To account for coarse fraction contained in the soil samples in SOC stock estimations, respective SOC horizon stocks (SOC<sub>stocki</sub>) were calculated in two steps following equations proposed by Poeplau et al. (2017). In a first

step, the so-called 'fine soil stock' for a respective horizon (j) is estimated (FSS<sub>j</sub>), (Eq. 2), where the mass<sub>fine soil</sub> is the mass of the total sample without the mass of the coarse fraction, volume<sub>sample</sub> is the total volume of the sample including the coarse fragments and depth<sub>j</sub> is the thickness of a respective horizon. In a second step SOC stocks are calculated for a respective horizon (SOC<sub>stock<sub>j</sub></sub>) (Eq. 3), where the SOC<sub>conc<sub>fine soil</sub></sub> is the OC concentration of the fine soil multiplied with the previously calculated fine soil stock FSS<sub>j</sub> (i.e. the total sample without the mass of the coarse fraction). Whole profile SOC stocks were calculated by summing all horizon specific SOC stocks per sampling site.

$$FSS_{j}\left(kg\,m^{-2}\right) = \frac{mass_{fine\,soil}\left(kg\right)}{volume_{sample}\left(m^{-3}\right)} \cdot depth_{j}\left(m\right) \tag{2}$$

$$SOC_{stock_j}(kg m^{-2}) = SOC_{conc_{fine\ soil}}(\%) \cdot FSS_j(kg C m^{-2})$$
(3)

#### 2.4.3 Calculation and mapping of geologic and climatic boundary conditions




To understand the generalizability of the results from this study to all European alpine grasslands, we analyzed the extent to which our geologic and climatic boundary conditions apply within QGis (3.40.5-Bratislava). This entailed the production of maps that show 1) the alpine grassland/meadow area within the extent of the European Alps, defined as the area included in the European Strategy for the Alpine Region (EUSALP), akin to the definition used by Marsoner et al. (2023), 2) lithologies/geologies underlying the EUSALP region. With these data, alpine grassland area coverage was calculated per lithology/geology class for those classes corresponding to those sampled within this study. These coverages were calculated based on: 1) all European countries (except Switzerland and Liechtenstein) that lay within the EUSALP region, 2) separately for Switzerland and Liechtenstein. Furthermore, the relative area covered by the MAT and MAP ranges of the study sites, i.e.  $2 \pm 1$  °C and  $1210 \pm 100$  mm was calculated.

Alpine grassland/meadow data was extracted from Marsoner et al. (2023). They created a detailed land use/landcover map for the areas included in the EUSALP, with a spatial resolution of up to 5 m and a temporal extent from 2015–2020. It was created by aggregating 15 high-resolution layers resulting in 65 land use/cover classes. The overall map accuracy was assessed at 88.8 %. Herein, natural alpine grassland was defined as being > 2000 m elevation. European lithology/geology coverage, at a resolution of 1:1'000'000 was taken from the European Geological Data Infrastructure (EGDI) (European Geological Data Infrastructure (EDGI), 2018). Swiss geology coverage was extracted from swisstopo ©, at a resolution of 1:500'000 (Federal Office of Topography swisstopo, 2012). MAP data (30 arc sec, ~ 1 km, based and a temporal extent from 1970–2000) was derived from WordClim Version 2.1 (Fick and Hijmans, 2017) and MAT data (30 arc sec, ~ 1 km, and a temporal extent from 1981–2010) from Climatologies at high resolution for the earth's land surface areas (CHELSA) (Karger et al., 2017, 2018).

#### 2.5 Statistical analyses






Data are presented as averages of related replicates with standard errors, unless specified otherwise. All statistical analyses were performed in R Studio (Version 2023.09.1+494). Please note that the respective sampling plots of Marl, that displayed a parent material geochemistry that did not correspond with the geochemical composition of a typical Marl, exhibiting markedly higher Si contents, were excluded from statistical analyses. One-way ANOVA was used to assess the main effect of site-specific parent materials on above- and belowground plant OC stocks as well as total SOC stocks. In each ANOVA, we tested the null hypothesis that there are no significant differences in OC stocks across the soils developed on different, site-specific parent materials. Tukey's HSD post-hoc tests were conducted to identify specific group differences. Statistical significance was set at a p-value < 0.05. We tested for normality using visual inspection such as OO- and scale-location plots. To analyze the effect of site-specific parent materials on aboveground plant organic carbon stocks, we employed a non-parametric alternative, due to the violation of the normality assumption required for ANOVA. Specifically, we conducted a Kruskal-Wallis test to assess the main effect of site-specific parent materials on aboveground plant organic carbon stocks. Then we performed Dunn's post hoc test to identify specific group differences. Spearman correlations were used to examine relationships between select soil biogeochemical variables, as this non-parametric method is appropriate for data that are non-normally distributed. A principal component analysis (PCA) was conducted to explore patterns and relationships among all measured bulk variables, SOC, and 'Depth', a variable capturing the lower depth of each horizon measured from the soil surface. All input variables were standardized using z-score normalization prior to analysis. The PCA was performed using the prcomp function of the R package 'stats' (R Core Team 2023), which applies singular value decomposition (SVD) to extract principal components (PCs) that capture the major axes of variation in the data. The resulting eigenvectors (loadings) describe the correlation between each variable and a given PC. We also calculated variable contributions to PC1 and PC2 as the sum of squared loadings across these components, expressed as percentages. These variable contributions served as visual indicators of each variable's relative importance in defining the PCA structure in a resulting biplot.

In connection to our research hypotheses that i) soils with greater soil fertility, enable better plant growth and potentially higher SOC stock accumulation and ii) soils with a more reactive mineral phase should contribute to greater SOC stabilization, we were interested in which set of parameters underlying these hypotheses could better predict SOC content in alpine grasslands. With this in mind, two different, hypothesis-driven SOC prediction models were built and compared to see if one model could outperform the other. For the soil fertility model C:N ratio, clay content, pH value and soil nutrient variables (i.e. bio-P, Mg, Ca) were selected as predictor variables. Soil C:N ratio was included because it reflects the stability/degradability of organic matter and how nutrients, especially N, will cycle through the soil (Havlin et al., 2013). Clay content was included, as their negative and variable charge give them a greater ability to bind to positively charged nutrient ions, contributing to soil fertility and nutrient retention (Kleber et al., 2015). Soil nutrient variables and pH were included as the overall concentration and availability of soil nutrients affects soil fertility (Havlin et al., 2013). For the soil mineralogy model organically-complexed and poorly crystalline Fe, Al, and Mn oxides as well as clay content were selected. These variables were included because

of their significant ability to stabilize OC through associations or organo-mineral complexes as a result of their large reactive surface area (Kögel-Knabner and Kleber, 2011; Kleber et al., 2015). Clay is included in both models to act as a proxy for nutrient retention and water holding capacity of the soil Yu et al. (2022), in the case of the soil fertility model, and as a proxy for SOC stabilization potential in the soil mineralogy model, as has been commonly done in SOC prediction models (Abramoff et al., 2021; Georgiou et al., 2022).




The number of predictor variables was kept < 10 due to limited observation availability (n = 42). Prior to predictive modelling, all predictor variables were z-score normalized to avoid any scale bias. A selection of linear and nonlinear regressions were computed, of which the nonlinear random forest regression was ultimately chosen as the most suitable model (Table B1). The maximum number of trees was constrained to 1000. The random forests' tuning parameter,  $m_{try}$ , the number of randomly selected predictors chosen at each split, was set to be one third of the number of total possible predictors (Kuhn and Johnson, 2013; Breiman, 2001). We used an adapted leave-one-out cross validation (i.e., leave-one-plot-out CV) to account for spatial autocollinearity within plots and thus to reduce overfitting (Yates et al., 2023). This cross validation iterates through all plots, while leaving out one plot per iteration and using its observations as test data, while the rest of the dataset was used as a training set, until all plots were used once as test data, on which we tested the prediction accuracy of our model. The random forest regressions were run using the caret package (Kuhn, 2008). For the predictive modelling of SOC content, only mineral horizons were included, as organic horizons have distinctly different SOC dynamics (Salomé et al., 2010). Hence, the Dolomite site's samples were excluded.

#### 3 Results

#### 3.1 Characterization of the soil profiles based on soil fertility and soil mineralogical properties

The widest C:N ratios in average aboveground plant biomass are those of the Gneiss and Greenschist sites (40.7 ± 2.0 vs. 60.8 ± 3.9, respectively) and the narrowest are at the Marl and Flysch sites (27.8 ± 5.4 vs. 25.9 ± 1.0, respectively) (Table 4). The widest C:N ratios in average belowground plant biomass can be found at the Gneiss and Greenschist sites (84.4 ± 12.7 vs. 75.1 ± 2.2, respectively) and the narrowest at the Marl and Flysch sites (33.8 ± 3.7 vs. 30.5 ± 2.2, respectively) again. Bulk soil C:N ratios reflect the trends from the above- and belowground biomass C:N ratios. Thus, the widest average bulk soil C:N ratios can be found at the Gneiss and Greenschist sites (16.1 ± 1.5 vs. 12.2 ± 0.2, respectively) and the lowest at the Marl and Flysch sites (11.2 ± 0.8 vs. 9.7 ± 0.4, respectively). The lowest pH average values can be found at the Gneiss (3.8 ± 0.1) and the highest at the Marl site (5.6 ± 0.8). Bio-P concentrations are highest at the Flysch site, ranging from 50.4–1.1 mg kg<sup>-1</sup> and lowest at the Greenschist site, ranging from 22.4–0.6 mg kg<sup>-1</sup>. However, there is no significant difference in bio-P concentrations across sites (p > 0.05). Marl and Greenschist's sites have the largest average CEC<sub>eff</sub> (25.0 ± 2.9 vs. 20.8 ± 1.3 cmol kg<sup>-1</sup>, respectively) and TRB (33.3 ± 46.2 vs. 20.8 ± 8.3 g kg<sup>-1</sup>, respectively). The Gneiss site has the lowest average CEC<sub>eff</sub> (9.9 ± 1.2 cmol kg<sup>-1</sup>) and TRB (8.9 ± 1.8 g kg<sup>-1</sup>).

Average Fe<sub>DCB</sub>/Fe<sub>tot</sub> ratios are highest at the Marl and Gneiss sites (0.72  $\pm$  0.22 vs. 0.68  $\pm$  0.18, respectively) and smallest at the Flysch site (0.38  $\pm$  0.08) (Table 4). Average sand content is highest (58.9  $\pm$  7.1 %) and clay content is lowest (3.8  $\pm$  0.7 %) at the Gneiss site. The largest average clay content is at the Marl site (13.7  $\pm$  5.8 %). The average sum of pedogenic oxides  $\Sigma$ Ped ox<sub>PP+AO</sub>) is highest at the Gneiss and Greenschist sites (18.0  $\pm$  1.6 vs. 18.4  $\pm$  1.6 g kg<sup>-1</sup>, respectively) and lowest at the Flysch site (8.7  $\pm$  1.7 g kg<sup>-1</sup>).

A PCA conducted on measured bulk soil biogeochemical variables, including SOC and depth, revealed clear clustering by parent material, along a pH–C:N gradient (Fig. S1). Depth was also shwon to structure observations between each site. The C:N ratio was found to align positively with exchangeable- and pyrophosphate-extractable Fe and Al, whereas pH associated positively with DCB-extractable (i.e. more crystalline) Fe and Al phases. Along this gradient, Gneiss-derived soils formed a clearly discernible group from the other parent materials. SOC is strongly associated with total N, sand content, and bioavailable P, while being negatively associated with depth, as expected. Contrarily, depth aligns positively with ammonium oxalate-extractable Fe and Al. Depth appears orthogonally to the pH-C:N gradient, suggesting that vertical depth-related changes in soils are structured independently of the major pH-C:N gradient.

**Table 4.** Average ± standard errors of C:N (-), pH (-), bio-P (mg kg<sup>-1</sup>), CEC<sub>eff</sub> (cmol kg<sup>-1</sup>), TRB (g kg<sup>-1</sup>), the sum of Al, Fe and Mn PP- and AO-extractable pedogenic oxides (∑Ped oxpp+AO) (g kg^-1), Felot/Fedoc (-), sand (%), clay (%) and coarse fraction (%) split into the following samples: aboveground biomass (ABM), belowground biomass (BBM) and individual soil horizons Oh, Ah, Bw, Cw.

| Parent material | Sample | C:N                             | Hd          | Bio-P         | CECeff         | TRB            | ∑Ped 0x <sub>PP+AO</sub> | Fedcb/Fetot     | Sand           | Clay           | Coarse fraction |
|-----------------|--------|---------------------------------|-------------|---------------|----------------|----------------|--------------------------|-----------------|----------------|----------------|-----------------|
| Dolomite        | ABM    | $31.4 \pm 1.8$                  |             |               | ı              | ı              |                          |                 | ı              |                |                 |
|                 | BBM    | $39.6\pm3.3$                    |             |               |                |                | 1                        |                 |                |                | 1               |
|                 | Oh     | $12.8\pm1.4$                    | $5.1\pm0.2$ | $54.4\pm12.3$ | $46.1 \pm 3.3$ | $19.7\pm1.5$   | $11.1\pm1.4$             | $0.79\pm0.02$   | $66.8\pm1.4$   | $1.3\pm0.2$    | $1.6\pm0.9$     |
| Marl            | ABM    | 27.8 ± 5.4                      | 1           | 1             | 1              | ı              |                          |                 | 1              | 1              | 1               |
|                 | BBM    | $33.8\pm3.7$                    |             |               |                |                | 1                        |                 |                |                | 1               |
|                 | Ah     | $12.5\pm0.6$                    | $5.3\pm0.4$ | $34\pm3.5$    | $32.9\pm2.9$   | $15.3\pm3.8$   | $11.1 \pm 3.3$           | $0.82 \pm 0.21$ | $73.8\pm5.3$   | $3.1\pm0.9$    | $6.9 \pm 5.5$   |
|                 | Bw     | $10.0\pm0.5$                    | $4.8\pm0.6$ | $2.3\pm0.4$   | $17.9\pm0.3$   | $16.9\pm0.5$   | $14.6\pm4.0$             | $0.70\pm0.01$   | $28.2\pm12.1$  | $15.7\pm5.7$   | $41.7 \pm 38.3$ |
|                 | Cw     | $11.0\pm1.9$                    | $6.6\pm0.2$ | $1.7\pm1.4$   | $24.1\pm0.3$   | $67.8\pm46.0$  | $7.6\pm0.6$              | $0.65\pm0.06$   | $11.1\pm0.9$   | $32.2 \pm 0.0$ | $37.1 \pm 17.8$ |
| Gneiss          | ABM    | $40.7 \pm 2.0$                  |             | ,             |                | ı              | ı                        |                 | ı              | 1              |                 |
|                 | BBM    | $84.4 \pm 12.8$                 | 1           |               |                |                |                          |                 |                |                | 1               |
|                 | Ah     | $17.5\pm0.9$                    | $3.7\pm0.1$ | $33.8\pm11.5$ | $15.6\pm0.9$   | $7.3\pm1.2$    | $12.3\pm1.2$             | $0.81 \pm 0.18$ | $74.7 \pm 6.1$ | $1.4\pm0.4$    | $16.1\pm0.6$    |
|                 | Bw     | $15.8\pm0.9$                    | $3.7\pm0.1$ | $3.2\pm0.3$   | $8.9\pm0.5$    | $8.6\pm0.9$    | $20.9 \pm 0.7$           | $0.67 \pm 0.03$ | $44.8\pm2.7$   | $4.9\pm0.5$    | $32.4\pm4.6$    |
|                 | Cw     | $15.1\pm0.8$                    | $4.0\pm0.0$ | $0.9\pm0.2$   | $5.1\pm0.7$    | $10.7\pm0.9$   | $21.9\pm0.7$             | $0.56\pm0.02$   | $57.1 \pm 2.5$ | $5\pm0.3$      | $46.6\pm2.1$    |
| Greenschist     | ABM    | $60.8 \pm 4.0$                  | -           | ı             | 1              | ı              | ı                        | 1               |                | ı              | 1               |
|                 | BBM    | $\textbf{75.1} \pm 11.6$        | 1           |               |                | 1              | 1                        |                 | 1              | 1              | 1               |
|                 | Ah     | $13.5\pm0.1$                    | $5.0\pm0.2$ | $22.4\pm3.3$  | $29.6 \pm 0.4$ | $40.3\pm3.7$   | $15.8\pm0.9$             | $0.56\pm0.02$   | $55.7 \pm 3.2$ | $4.6\pm0.5$    | $4.7\pm0.6$     |
|                 | Bw     | $11.2\pm0.2$                    | $4.7\pm0.2$ | $3.8\pm0.5$   | $17.8\pm0.8$   | $47.4\pm4.8$   | $20.5\pm0.8$             | $0.55\pm0.03$   | $22.1 \pm 3.0$ | $12.1\pm1.1$   | $9.9\pm0.9$     |
|                 | Cw     | $11.9\pm0.1$                    | $5.0\pm0.1$ | $0.6\pm0.1$   | $15.1\pm0.9$   | $82.3 \pm 5.7$ | $17.7\pm1.1$             | $0.35\pm0.03$   | $19.8\pm2.0$   | $12.9 \pm 0.6$ | $52.5\pm2.6$    |
| Flysch          | ABM    | $25.9 \pm 1.0$                  | ı           | 1             | 1              | ī              | ī                        | 1               | ı              | ī              | ,               |
|                 | BBM    | $30.5\pm2.1$                    | ı           | 1             | 1              | 1              |                          | 1               | 1              | ı              | 1               |
|                 | Ah     | $12.0\pm0.4$                    | $5.1\pm0.2$ | $50.4\pm6.6$  | $22.7\pm3.2$   | $15.4\pm1.7$   | $7.5\pm0.8$              | $0.38 \pm 0.04$ | $62.5\pm3.4$   | $2.4\pm0.4$    | $9.8\pm1.2$     |
|                 | Bw     | $9.5\pm0.6$                     | $4.6\pm0.3$ | $6.1\pm1.8$   | $8.8\pm1.2$    | $14.3\pm1.6$   | $9.7\pm1.2$              | $0.39 \pm 0.06$ | $32.5\pm3.5$   | $6.0\pm0.5$    | $37.9 \pm 3.1$  |
|                 | Cw     | $\textbf{7.5} \pm \textbf{0.1}$ | $5.1\pm0.4$ | $1.1\pm0.2$   | $6.9\pm0.5$    | $15.5\pm1.8$   | $8.8\pm1.0$              | $0.37 \pm 0.04$ | $24.9\pm1.9$   | $7.1\pm0.5$    | $44.8 \pm 3.4$  |
|                 |        |                                 |             |               |                |                |                          |                 |                |                |                 |

#### 3.2 Above- and belowground biomass organic carbon stocks and soil organic carbon stocks

Across all sites, the significantly largest ABM OC stock can be found at the Gneiss site, amounting to  $754 \pm 144$  g C m<sup>-2</sup> (Fig. 3 [a]). Of which an approximate 80 % consists of woody plant material. For all other sites the woody biomass presence is either zero or < 20 %. The Dolomite and Marl sites show the lowest aboveground biomass OC stocks ( $147 \pm 12$  and  $140 \pm 31$  g C m<sup>-2</sup> respectively), while not varying significantly from that of the Flysch and Greenschist sites. The highest belowground biomass OC stocks can be found at the Flysch and Marl sites ( $91 \pm 10$  and  $97.3 \pm 24$  g C m<sup>-2</sup> respectively), which are approximately three times larger than those of the other sites. These two sites also show a more constrained average above-to belowground biomass OC stock ratios of ~1.6–2.5, compared to the other sites with ratios of 4 up until 22 for the Gneiss site.

The largest whole profile SOC stocks can be found on the Gneiss site, closely followed by those of the Marl site, that showed no statistically significant difference (23.2  $\pm$  1.9 and 21.3  $\pm$  4.3 kg C m<sup>-2</sup> respectively) (Fig. 3 [b]). The smallest whole profile SOC stocks can be found on the Flysch and Dolomite sites (8.1  $\pm$  1.2 and 9.5  $\pm$  1.5 kg C m<sup>-2</sup> respectively), which are significantly smaller than the stocks of the other three sites. Across all sites, the largest horizon-specific stocks are contained within the Bw horizons, that contribute an approximate 74–87 % to whole profile SOC stocks. The smallest horizon specific SOC stocks are those of the Cw horizons.

#### 3.3 F<sup>14</sup>C and the relative contribution of soil fractions to bulk soil organic carbon

Across all sites there is a depth/horizon-dependent decrease in  $F^{14}C$  values (Fig. 4 [a]). The overall smallest  $F^{14}C$  values are found at the Marl and Gneiss sites. Hereby, the Marl site displays a smaller value for the Cw horizon than the Gneiss site (0.77 vs. 0.82, respectively) but a slightly higher value for the Bw horizon compared to the Gneiss site (0.97 vs. 0.95, respectively). The greatest overall  $F^{14}C$  values for all soil horizons can be found for the Greenschist site. Ranging from 1.06–0.88 from the Ah to the Cw horizon. All sites' Ah horizons are modern in terms of  $F^{14}C$ , with values > 1. However, Marl site's Ah horizon seems to include slightly greater amounts of more persistent C, leading to a slightly smaller  $F^{14}C$  value of 1.04, compared to the value of 1.06 for the other sites. The only other horizon that is deemed modern is the Bw horizon of the Greenschist site.


The importance of a fraction's relative contribution to bulk SOC (%), i.e., how much the C contained within a fraction contributes to bulk SOC, varies within each site's soil horizons and across all sites (Fig. 4 [b]). A first overarching trend across all sites can be found for the relative contribution of POM to bulk SOC (%), which increases with increasing bulk SOC (%). The POM fraction's relative contribution to bulk SOC varies the most of all fractions and ranges between 1–73 % (Fig. 4 [b], POM). The s+c fraction's contribution to bulk SOC decreases rather linearly with increasing bulk SOC. It therefore displays a trend opposite to that of the POM fraction, but increases with depth. The s+c fraction's overall contribution ranges from 3–44 % to bulk SOC (%) (Fig. 4 [b], s+c). In contrast, the relative contribution of the MA fraction to bulk SOC does not show a clear pattern with varying bulk SOC content. This fraction's relative contribution to bulk SOC is > 25 % across all soil horizons of all sites, with most values lying within 45–75 % (Fig. 4, MA).

Figure 3. (a) Average  $\pm$  standard errors of total above- and belowground plant biomass OC stocks (g C m<sup>-2</sup>) across all sites. Aboveground biomass values are depicted as > 0 and belowground as < 0 for illustrative purposes. The relative amount of woody aboveground plant biomass is represented by the black crosshatch pattern within the bars, and the solid part of the bars represents the relative amount of herbaceous/non woody biomass. (b) Average total SOC stocks (kg C m<sup>-2</sup>)  $\pm$  standard errors, split into individual horizon SOC stocks, across all sites. Same respective lowercase, or uppercase letters denote total above-/belowground plant OC or whole profile SOC stocks, that are not statistically significant from one another.

#### 3.4 Soil organic carbon model predictions with soil fertility versus soil mineralogical parameters




Both the soil fertility and the soil mineralogy model's predictions of bulk SOC content (%) aligned well with the observed SOC values (Fig. 5 [a] and [c]). The soil fertility model slightly outperforms the soil mineralogy model with an  $R^2$  of 0.91 and an RMSE of 2.02, compared to an  $R^2$  of 0.84 and an RMSE of 2.52. Bio-P has the highest relative importance for the soil fertility model, with a value of 20 % and has a significant positive correlation coefficient of 0.86 with bulk SOC (Fig. B1). Clay is among the most important predictor variables for both models, with a relative variable importance of 17.7 and 24.2 % for the soil fertility and soil mineralogy model, respectively (Fig. 5 [b] and [d]). Clay shows a significant negative correlation coefficient of -0.77 with SOC (Fig. B1, Fig. B2). The soil fertility model's predictor variables are all significantly, positively correlated with SOC content except for clay, exchangeable K ( $K_{ex}$ ) and pH, which are negatively correlated. The soil mineralogy models' organically-complexed Fe<sub>PP</sub>, Al<sub>PP</sub> and Mn<sub>PP</sub> oxides are all positively correlated with SOC content, while the poorly crystalline Fe<sub>AO</sub>, Al<sub>AO</sub> and Mn<sub>AO</sub> oxides correlate negatively with SOC with varying degrees of significance (Fig. B2).

Figure 4. (a)  $F^{14}C$  of all soil horizons and sites. Note that all values > 1 are defined as modern. (b) Average bulk SOC (%)  $\pm$  standard errors vs. the average relative contributions of the three examined fractions POM, MA and s+c to bulk SOC (%)  $\pm$  standard errors are shown across all sites and their individual horizons. Observations from the Dolomite site were excluded from this figure as its Oh horizon is organic and showed signs of POM contamination in the MA fraction.

#### 4 Discussion

#### 4.1 Largest biomass organic carbon stocks found on low fertility soil

Considering the sampling sites' ranges of soil fertility variables, the Marl site shows the best, and the Gneiss site the worst, plant growth conditions. The Marl site's pH values are within a range, where essential plant micronutrients and phosphorus are plant-available (Barrow et al., 2020; Blume et al., 2016). The given pH range, together with the Marl site's bulk soil C:N values, support microbial processing and hence the decomposition of OM to nutrients and energy for plant and microbial growth (Brust, 2019; Barrow et al., 2020; Currie, 2003; Myrold and Bottomley, 2008). The Marl site also has the overall highest CEC and TRB values which support nutrient retention in the soil (Blume et al., 2016). In terms of soil mineralogy variables, the Marl and Gneiss sites show the best SOC stabilization potential conditions, by having the highest respective clay content and pedogenic oxide concentrations. With the second-highest overall pedogenic oxide concentrations of all sites, Greenschist also possesses good conditions for potential SOC stabilization. The Flysch site has the overall worst combination of plant growth and SOC stabilization conditions. Possessing the lowest pedogenic oxide concentrations and simultaneously having a relatively low clay content.

**Figure 5.** (a) Predicted bulk SOC (%) values *vs.* measured/observed bulk SOC (%) values of the soil fertility model. (b) Predicted bulk SOC (%) values *vs.* measured/observed bulk SOC (%) values of the soil mineralogy model. (c) Relative importance (%) of the model predictor variables included in the soil mineralogy model. (d) Relative importance (%) of the model predictor variables included in the soil fertility model. The dashed line represents a 1:1 regression line.

Against the posited hypothesis that nutrient-rich soils with a high mineral weatherability, such as those of the Marl or Greenschist sites, would display the largest plant biomass OC stocks, the highest total plant biomass OC stocks were found at the Gneiss site (Fig. 3 [a]). These high stocks are mostly attributable to the Gneiss site's ABM stocks which contribute approximately 95  $\pm$  25 % to total plant biomass OC stocks. The soil conditions present at the Gneiss site, such as an acidic pH range and sandy texture, providing good water drainage, benefit the encroachment and growth of shrubs of the Vaccinium genus (Chen et al., 2019; Nestby et al., 2011; Ritchie, 1956). The growth of these shrubs contribute to the very high percentage of woody biomass and large C:N ratios at this site (Fig. 3 [a], Table 4). Furthermore, shrub encroachment in sandy-textured soils can have beneficial effects on SOC, given the higher biomass input (Li et al., 2016). The greater aboveground plant biomass of this Vaccinium-dominated site, and its slightly elevated composite plant biomass C concentration (46.3 vs. 43.9–44.1 % compared to all sites except Greenschist), result in the observed large ABM OC stocks. Despite the Gneiss site having some of the lowest measured CEC<sub>eff</sub> and TRB values (Table 4), we infer that the Vaccinium may be able to support their relatively large ABM through N fixation and facilitated uptake of P by mycorrhizal roots (Jiang et al., 2024; Vohník and Réblová, 2023; Daghino et al., 2022; Püschel et al., 2021). Recently, Zhao et al. (2023) also showed that shrub encroachment in alpine grasslands of the Tibetan Plateau led to increased aboveground biomass compared to non-encroached sites. Despite its large ABM, the Gneiss site has significantly smaller BBM OC stocks than those of the Marl, Flysch and also Greenschist sites (Fig. 3 [a]). The Dolomite site also exhibits low plant biomass OC stocks, which is typical for the little developed, nutrient-poor soil (Pignatti



and Pignatti, 2014). In contrast to ABM OC stocks, the differences in BBM OC stocks appear to more closely follow differences in CEC<sub>eff</sub>, TRB and clay, which all contribute to soil fertility. The differences in these BBM OC stocks are also reflected by the relative rooting depth differences within the different sites' soil profiles (Fig. 2). Overall, the above- and belowground biomass of all sites measured in the context of this study (240–2440 g m<sup>-2</sup> and 14–320 g m $^{-2}$ , for above- and belowground biomass, respectively) lie within a similar range of above- and belowground biomass reported for a medium alpine altitudes under closed alpine vegetation (100–1500 g m $^{-2}$  and *ca.* 500–700 g m $^{-2}$ , for above- and belowground biomass, respectively) (Leifeld et al., 2009; Hitz et al., 2001).





In summary, the importance of soil fertility variables such as CEC<sub>eff</sub>, TRB and clay for biomass production seem to have a more pronounced effect on BBM OC stocks than ABM OC stocks. Our data shows that ABM OC stocks can become large despite soils not being nutrient rich or displaying high CEC<sub>eff</sub> values, yet displaying properties that favour the growth of specific, site-adapted plant functional types.

# 4.2 Magnitude of biomass organic carbon stocks not necessarily linked to magnitude of soil organic carbon stocks - importance of soil stabilization mechanisms

The highest overall SOC stocks were found at the Gneiss and Marl sites (Fig. 3 [b]). Since the Gneiss site harbours the highest plant biomass OC stocks, our hypothesis that high biomass OC stocks may lead to, or coincide with, the largest SOC stocks therefore seems true at first glance. Increased SOC stocks under shrub-encroached, alpine grasslands were also found on the Tibetan Plateau (Zhao et al., 2023). This shrub-induced increase in SOC stocks is presumably linked to the high sand content found at the Gneiss site, which provided ideal conditions, such as an acidic pH and good water drainage, for the encroachment of shrubs of the *Vaccinium* genus. A meta-analysis of worldwide shrub-encroached grasslands showed that resulting SOC content changes were soil texture dependent, with decreases in silty and clay soils and increases in sandy soils (Li et al., 2016). However, the meta-analysis also shows that SOC content changes following encroachment were genera-dependent. Lastly, they found that the main drivers behind SOC content gain or losses with shrub encroachment were the relative differences in productivity between shrub-encroached vs. non shrub-encroached grasslands. However, the Marl site possesses a much smaller total plant biomass OC stock but still harbours comparable SOC stocks.

Overall magnitude of total plant biomass does not necessarily determine or coincide with SOC accrual. The potential of soil stabilization mechanisms must be taken in consideration additionally, as we hypothesized. This becomes all the more apparent considering the Flysch site which has a greater ABM and comparable BBM OC stocks to the Marl site. Yet, the Flysch site's SOC stock is only approximately one third the size of the Marl and Gneiss site's SOC stocks. These findings contradict those of Nie et al. (2023), that found a significant correlation, while moderate in strength, between ABM and BBM on SOC stocks across alpine meadows of the Tibetan Plateau. The Dolomite site's small SOC stocks are likely caused by the absence of mineral horizon formation, limiting the accumulation of greater SOC stocks at this site. The lack of mineral horizons are the result of the high solubility/weatherability of the Dolomite site's parent material (Brady and Weil, 2008). Nevertheless, the Dolomite

site's thick Oh horizon still holds SOC stocks comparable to whole profile SOC stocks of the Flysch site. Despite the Flysch site having developed a deeper soil profile including mineral horizons (Fig. 2, Fig. 3 [b]). On the whole, the SOC stocks of all sites we report here (8–23 kg C m<sup>-2</sup>) lie within a similar range of SOC stocks reported for medium alpine altitudes under closed alpine vegetation (4–22 kg C m<sup>-2</sup>) (Webber and May, 1977; Canedoli et al., 2020; Leifeld et al., 2009).








Our second hypothesis stated that pedogenic oxides and not the amount of clay would ultimately coincide with greater SOC stocks. Supportive of this hypothesis is the positive relationship between Fe<sub>PP+AO</sub> pedogenic oxide concentrations and horizondependent SOC stocks, with the highest Fepp+AO pedogenic oxide concentrations coinciding with the highest SOC stocks in the Bw horizons (data not shown). In comparison, the relationship between Alpp+AO and Mnpp+AO pedogenic oxides and SOC stocks (data not shown) and SOC content (Fig. B2) is less evident. Past work has reported varying importance of Alpp-AO compared to Fe<sub>PP+AO</sub>. With some studies reporting a stronger relationship between Al<sub>AO</sub> and SOC than Fe<sub>AO</sub> and SOC (Hall and Thompson, 2022; Percival et al., 2000) and other work demonstrating the relevance of both AlAO and FeAO to SOC stabilization in cold and wet ecosystem (Yu et al., 2021). Nevertheless, across all sites, the highest sum of FepphaO and AlpphaO pedogenic oxide concentrations are located at the Gneiss and Greenschist sites, as a result of their mafic parent materials. The co-occurrence of high Fepphao and Alpphao pedogenic oxide concentrations and the largest SOC stocks at the Gneiss site, underline the importance of pedogenic oxides in SOC stabilization there, especially since the Gneiss site has a very low clay content (Table 4. In contrast, the Marl site, which holds similarly large SOC stocks as the Gneiss site, has the lowest Fepphan concentrations and smaller Alpp+AO concentrations. Concomitantly, the Marl site has the highest clay content of all sites and may thus be of higher importance at this site for SOC stabilization through complexation and adsorption on reactive clay surfaces but also by promoting the production of aggregates (Six et al., 2002; Lützow et al., 2006). The larger importance of clay-mediated SOC stabilization processes at the Marl site is supported by a greater contribution of this site's s+c fraction C to bulk SOC in the Bw horizon compared to that of the other sites (Fig. 4, [b]).

As a result of the SOC stock calculation methodology applied to horizon-specific sampling and analyses, instead of fixed depth increments, the magnitude of horizon-specific and whole profile SOC stocks depend on the thickness of individual soil horizons and their respective OC contents (see Sect. 2.4.2). Thus, because thick horizons (such as Bw) can harbor more C as part of the profile stock, even if C concentrations are lower than in thinner (topsoil O or A) horizons. Thus, if SOC stocks were normalized to the same depth increment thickness, the trends reported here would potentially change. Furthermore, SOC stock estimates can vary significantly depending on the calculation method applied. For example, Poeplau et al. (2017) show that discrepancies in estimated SOC stocks between methods increase with increasing rock fragment content. In soils with rock fragments comprising > 30 vol%, they revealed that SOC stocks may be overestimated by as much as 100 %. Thus, as our sampling took place in steep terrain with strongly varying rock fragment contributions of  $1.6 \pm 0.9$  in the Dolomite's Oh, up to  $52.5 \pm 2.6$  for Greenschist's Cw horizon (Table 4), we calculated SOC stocks by taking these varying contributions into account, as to not significantly overestimate SOC stocks by applying the calculation methodology as suggested by Poeplau et al. (2017) (Sect. 2.4.2). In addition to uncertainties related to the SOC stock calculation methodology, spatial heterogeneity

of SOC further introduces uncertainty. Due to the formation of microenvironments, soil-forming processes can vary at very small scales (Körner, 2003a; Kemppinen et al., 2024). We aimed to minimize this effect by collecting composite samples which consisted of 10 individual samples per horizon and plot (Sect. 2.3.1). Further, the sites of this study were chosen to cover a broad range of some of the most important geologies underlying European alpine grasslands (Fig. S3, S4). The wide range in mineralogy, texture and formation of the chosen parent materials represent  $\sim 80$  % of the geologies in Switzerland and Liechtenstein and  $\sim 40$ –60 % of geologies in the rest of the European alps, underlying alpine grasslands (Tables S1, S2). One important geology that was not included in this work is quartzite. The MAT and MAP ranges of the sites included in this study only cover  $\sim 11.4$  and  $\sim 24.3$  % of all european alpine grasslands > 2000 m.a.s.l. respectively (Table S3). To conclude, even with coarse fraction–adjusted SOC stock calculation methods and careful site selection, composite sampling–inherent uncertainties associated with spatial heterogeneity of SOC will remain part of studies that estimate SOC stocks at a regional scale.

In summary, the formation of organo-metal complexes and association of OC with poorly crystalline Fe and Al oxides and short-range order (SRO) minerals pose an important mechanism contributing to overall SOC stocks. Especially, in soils developed on siliceous and mafic rocks, rich in Fe and Al, such as at the Gneiss or Greenschist sites. In soils with higher clay content and lower concentrations of Fe and Al pedogenic oxides, such as the Marl site, clay still contributes significantly to SOC stabilization.

#### 4.3 SOC persistence not explicable by soil fraction differences






The Marl and Gneiss sites hold the most persistent SOC of all sites in their subsoil horizons (Bw, Cw), when excluding the Dolomite site's single observation (Fig. 4 [a]). However, Dolomite's Oh horizon's F<sup>14</sup>C value appears to be smaller than the F<sup>14</sup>C values of all other site's Ah and Bw horizons, despite constituting an organic layer. We assume that the Oh horizon formed here does not decompose due to reduced turnover, caused by the cold temperatures and harsh environment of the alpine environment (Körner, 2003a). The produced organic matter presumably accumulates at this site because there are no subsoil layers it could be incorporated into, as these did not form on the Dolomite. The reduced decomposition is also reflected in the overall thickness of the layer (12.9  $\pm$  2.2 cm), which is nearly double as thick as all other sites' Ah horizons. Additionally, the calculated weathering proxy, Fe<sub>DCB</sub>/Fe<sub>tot</sub>, is highest at the Dolomite site, indicating that soils developed on Dolomite are highly weathered (2). The soils of the Gneiss and Marl sites have undergone stronger weathering, in terms of Fe<sub>DCB</sub>/Fe<sub>tot</sub> (Fig. 2, Table 4). In the case of Marl, a sedimentary rock, geogenic OC may be present that could additionally contribute to making the in situ SOC appear more persistent than if it were derived solely from biological inputs and pedogenic processes. Kalks et al. (2021) found that geogenic OC influence is of particular importance in young soils on terrestrial sediments with comparatively low amounts of OC. Hence the impact of these potential inputs would be largest in the subsoil, close to the weathering front. As geogenic OC was not specifically quantified in this study, its presence cannot definitely be ruled out in any of the sampled soils formed on sedimentary rocks. Nevertheless, the soils formed on sedimentary rocks do not have clear, greater contributions of MA or s+c to bulk SOC in comparison to all other sites, except for the Marl site's Bw horizon. Therefore our general hypothesis, that soils with more persistent SOC would have developed increased mineral-related stabilization mechanisms, compared to soils with less persistent SOC, must be rejected for most sites. The Marl site's Bw horizon marks a exception with a ≥ 1.5 time higher contribution of s+c to bulk SOC (%) compared to all other sites' Bw horizons, which show a relatively similar relative contribution of s+c to bulk SOC. With regard to the relative contribution of MA to bulk SOC, however, the Marl site's Bw horizon shows values similar to those of the other sites (Fig. 4 [b]). The higher SOC persistence of the Gneiss and Marl sites' subsoils, taken together with the importance of MA and s+c fraction's contribution to bulk SOC in these sites' Bw horizons, suggest that the majority of these sites' SOC stocks are rather stabilization- than input-driven. In comparison, the Greenschist site, which has the largest F<sup>14</sup>C values across all horizons of all sites including modern values for its Bw horizon, seems to be more input-driven with a potentially higher turnover of OC. A study on a 2200 km grassland transect on the Tibetan Plateau also found that topsoil (0–10 cm) Δ<sup>14</sup>C values were plant OC input controlled compared to subsoil (30–50 cm) Δ<sup>14</sup>C values, which depended on mineral stabilization mechanisms (Chen et al., 2021).

In contrast to their subsoils, the modern Ah horizons of the Gneiss and Marl sites show a much larger contribution of POM to the Ah horizons' bulk SOC (> 50 %). The absolute and relative importance of POM to the Ah horizon of the Gneiss site is presumably linked to the large, woody-biomass-dominated ABM OC stock, and thus potential input into the soil (Fig. 4, Table A1). The wide C:N ratio of the Gneiss site's ABM may also lead to reduced decomposition, and thus accumulation, of POM in the soil (Liang et al., 2017). At the Marl site the significance of POM to bulk SOC may result from its large production of belowground biomass OC (Fig. 3 [a]). For all other horizons no clear differences can be seen regarding relative fraction contributions to bulk SOC across the sites. The only clear trend in relative fraction contributions to bulk SOC is visible with increasing soil depth, where the contribution of s+c increases while that of POM diminishes. This finding further supports the initial hypothesis that clay as a particle size alone is less important a contributing factor to SOC stocks across all sites, except the Marl site.

We show potential differences in the importance of SOC stabilization mechanisms, with the MA soil fraction contributing ≥ 50 % to bulk SOC in the majority of cases, across all sites (Fig. 4, MA). With the exception of the Ah horizon at the Gneiss and Marl sites. The importance of the MA formation for SOC stabilization is also reflected by it contributing the most, out of all analyzed fractions, to bulk SOC within Bw horizons. The overall thickest horizons, with the largest relative contribution to whole profile SOC stocks, except for the Dolomite site (Fig. 2, Fig. 3 [b]). Consistent with these findings, (Wasner et al., 2024) examined a geoclimatic gradient across a diverse range of grassland topsoils showed that stable microaggregates were the biggest contributors to bulk SOC in C-rich soils. With both the SOC quantity in free silt and clay and stable microaggregates fractions positively correlating to pedogenic oxide contents and texture. Further evidence from (Lehndorff et al., 2021), which examined the spatial organization of microaggregates, demonstrates that greater OC concentrations are found within the microaggregate fractions compared to bulk SOC in a sandy and a loamy Luvisol. They also found a systemic increase in iron, clay and silicate mineral phases in the microaggregates formed on the clay-richer Luvisol, indicating that inherent soil mineralogy is reflected in the composition of microaggregates. Their data supports the notion that OC forms the core for MA formation and that pedogenic iron aids aggregation processes in the soil by acting as a cementing agent (Campo et al., 2014:

Totsche et al., 2018). The strong relevance of Fe for aggregation processes is also reflected in significant correlations of reactive Fe pedogenic oxides ( $Fe_{PP+AO}$ ) and total Fe pedogenic oxides ( $Fe_{PP+AO+DCB}$ ) with the MA mass (wt%) within the sites of this study (Spearman correlation coefficients of 0.53 and 0.48 respectively, with p < 0.05, Fig. S2). By contrast, Al and Mn pedogenic oxides showed no significant relationships (data not shown). We attribute this to low Mn pedogenic oxide concentrations and site pH values (4.2–6.8, except Granite at 3.8), which are above the range where Al becomes highly mobilized.






The interpretation of the relative importance and influence of soil fractions on SOC stabilization, is the result of the choice of the fractionation scheme. Leuthold et al. (2024) compared size-, density-, and combined-based methods across U.S. agricultural soils, to identify chemical similarities and differences across the methodologically-defined, and -derived, fractions. They found that MAOM fractions (comparable to the s+c fraction in this study), were were consistent in terms of their spectral, isotopic and chemical characteristics. Their MAOM C concentrations were similar across fractionation methods, but slightly higher in one-step methods. POM C varied widely across combined methods, compared to one-step methods. The fractionation method applied in this study was similar in nature to the first combined size and density fractionation methods examined by Leuthold et al. (2024). However, the MA and s+c fractions in our study were defined solely by size, following the physical breaking up of macroaggregates, rather than by size and density. Thus, considering the results of Leuthold et al. (2024), our study may overestimate POM C concentrations, compared to values that would have been obtained with a different fractionation methodology. Finally, depending on the fractionation protocol applied, no CHAOM or MA fraction is isolated. This further leads to a different importance attributed to the extracted fraction's relative C contributions to bulk SOC.

In summary, though the most weathered soils of the Gneiss and Marl sites display the most persistent bulk OC values, they did not show a larger contribution of mineral-stabilized C to bulk SOC, compared to all other sites. These two sites also correspond to the largest SOC stocks, that appear to be rather stabilization than input-driven. Further, across all sites and horizons a large contribution of MA was found to contribute to bulk SOC, that appear to be supported by the presence of reactive, pedogenic Fe phases. These findings underline the importance of aggregation processes for SOC stabilization in alpine soils developed on different parent materials.

#### 600 4.4 SOC content predicted equally well with soil fertility and soil mineralogical variables

Both SOC prediction models performed similarly well, despite portraying two different sets of drivers (Fig. 5). We interpret this finding as follows- that both soil fertility as well as soil mineralogical parameters are equally important in determining SOC content and that both sets of parameters can ultimately contribute to SOC accrual. In both models clay appears as one of the most important predictor variables of SOC content, despite recent work reporting other variables as better predictors of SOC content. Such as exchangeable Ca in water-limited soils and Fe and Al oxy(hydr-)oxides in wet, acidic soils (Rasmussen et al., 2018; von Fromm et al., 2021). We assume that on the site-level one model may outperform the other, depending on which set of parameters ultimately matter more at a specific site. For example, we assume that the soil fertility model could more accurately predict SOC of sites that show higher values of soil fertility-enhancing parameters, e.g. TRB, bio-P, clay and

simultaneously possess smaller values of properties that would enable a greater mineralogical stabilization potential for SOC, e.g. pedogenic oxides. This may be the case for the Flysch site's soil (Fig. 2, Table 4, Table A1). We therefore also assume that the soil mineralogy model could more accurately predict SOC of more strongly weathered soils, such as those of the Marl or Gneiss site, that have developed properties as mentioned above, enabling greater mineralogical stabilization potentials. However, due to the limited number of observations, we cannot confirm which of the two models leads to more accurate SOC predictions for each respective site. To assess the validity of our aforementioned assumptions, a larger number of observations would be required than those provided in this study.

In summary, soil fertility as well as soil mineralogical parameters performed equally well for predicting SOC content and can therefore ultimately both contribute to SOC accrual in these shallow to moderately deep, developing alpine soils.

#### 5 Conclusions



This study explored how parent material geochemistry affects soil properties and, in turn, influences plant biomass OC and SOC accumulation, through differences in resulting soil stabilization potentials and soil fertility in European alpine grasslands. Contrary to expectations, neither above- nor belowground plant biomass OC stocks were primary elements shaping whole profile SOC stocks. Instead, our data shows that more weathered soils developed deeper profiles and accumulated substantially greater SOC stocks, largely due to enhanced stabilization potential. Hereby, sites rich in Fe and Al pedogenic oxides facilitated 625 increased organo-metal complexation and occlusion of SOC by microaggregates. While clay had a relatively minor role in most soils, it was a notable contributor to SOC stock accrual at the site with the highest clay content and most persistent SOC. Across all study sites, microaggregates emerged as the most important universal SOC-stabilizing phase, contributing > 50 % to bulk SOC in most soils. Thus, we propose that future research should investigate how pH, oxy(hydr-)oxides and plant biomass inputs influence microaggregate formation, composition and stability across alpine soils of varying weathering stages. Finally, 630 by applying a random forest model, we demonstrated that SOC content in the examined alpine soils could be predicted equally well by a set of soil fertility or mineralogical variables- underscoring the dual importance of both soil fertility and mineralogy in determining SOC.

Code and data availability. Datasets used for this publication are available on Zenodo (https://doi.org/10.5281/zenodo.15282598, Maier et al., (pre publication version) as are the R code scripts (https://doi.org/10.5281/zenodo.17195820, Maier et al.

#### 635 Appendix A

**Table A1.** Overview of the results from the soil fractionation procedure. All results are reported as averaged values  $\pm$  the standard deviation. Fractionation mass loss shows how much sample material was lost during the fractionation procedure relative to the amount of initially weighed-in bulk soil, where positive values indicate a net mass loss and negative values indicate a supposed mass increase post-fractionation. The fraction percentages of POM, MA, and s + c are related to the total soil present after soil fractionation.

| Parent material | Horizon | Fractionation mass loss (%) | <b>POM</b> (%)  | MA (%)          | s + c (%)       |
|-----------------|---------|-----------------------------|-----------------|-----------------|-----------------|
| Dolomite        | Oh      | $6.9\pm1.4$                 | $32.6 \pm 11.6$ | $54.0 \pm 11.6$ | $13.5 \pm 0.7$  |
|                 | Ah      | $4.6 \pm 0.5$               | $38.9 \pm 27.4$ | $45.3 \pm 18.7$ | $15.8 \pm 8.6$  |
| M 1             | Bw      | $2.5\pm1.0$                 | $30.2 \pm 9.1$  | $41.2\pm11.5$   | $28.5 \pm 4.1$  |
| Marl            | Cw      | $0.8 \pm 0.9$               | $17.1\pm3.9$    | $29.8 \pm 16.1$ | $53.2 \pm 19.6$ |
|                 | Ah      | $3.0 \pm 1.8$               | $57.5 \pm 18.4$ | $37.6 \pm 15.8$ | $4.9 \pm 9.8$   |
| Gneiss          | Bw      | $2.2\pm0.7$                 | $37.3 \pm 7.8$  | $52.3 \pm 5.0$  | $10.4 \pm 7.3$  |
| Gneiss          | Cw      | $2.8\pm1.0$                 | $38.6 \pm 0.8$  | $46.9 \pm 2.6$  | $14.5\pm8.9$    |
|                 | Ah      | $4.2\pm2.3$                 | $19.6 \pm 1.1$  | $57.6 \pm 4.8$  | $22.8 \pm 3.7$  |
| Greenschist     | Bw      | $2.3\pm0.7$                 | $14.2 \pm 4.5$  | $61.3 \pm 5.0$  | $24.5 \pm 3.9$  |
| Greenschist     | Cw      | $1.1\pm0.9$                 | $30.3 \pm 4.3$  | $40.7\pm2.4$    | $29.0 \pm 2.7$  |
|                 | Ah      | $2.2\pm1.4$                 | $32.9 \pm 3.1$  | $49.4 \pm 3.0$  | $17.8 \pm 3.5$  |
| Elwach          | Bw      | $-8.2 \pm 15.9$             | $32.7 \pm 8.9$  | $44.6\pm7.7$    | $22.7 \pm 8.2$  |
| Flysch          | Cw      | $-0.1 \pm 0.5$              | $35.2 \pm 1.6$  | $37.1\pm5.5$    | $27.7 \pm 3.9$  |

### Appendix B

**Table B1.** Overview of the linear regression, least-angle regression, elastic net regression and random forest model performance metrics ( $R^2$ , RMSE). All results are reported as averaged values of 100 model runs.

| Model                  | Soil fertility variables |      | Soil mineralogy variables |      |  |
|------------------------|--------------------------|------|---------------------------|------|--|
|                        | $R^2$                    | RMSE | $R^2$                     | RMSE |  |
| Linear regression      | 0.90                     | 2.05 | 0.72                      | 2.96 |  |
| Least-angle regression | 0.88                     | 2.09 | 0.70                      | 3.06 |  |
| Elastic net regression | 0.89                     | 2.02 | 0.70                      | 3.04 |  |
| Random forest          | 0.89                     | 2.07 | 0.81                      | 2.56 |  |

Figure B1. In the left half of the matrix, scatterplots show the relationship between all soil fertility model predictor variables and SOC. Observations are shown considering individual parent materials and horizons, corresponding to different respective colours and shapes. Where Ah are squares, Bw diamonds and Cw triangles. Diagonally, density plots show observation frequencies for respective predictor variables are displayed and split into individual parent materials, according to different allocated colours. In the right half of the matrix, spearman correlations coefficients for all soil fertility model predictor variables and SOC, considering all observations (Overall Corr) and individual parent materials, corresponding to different respective colours. Note that "\*\*\*", "\*\*", "\*", " " indicate p-values of 

Figure B2. In the left half of the matrix, scatterplots show the relationship between all soil mineralogy model predictor variables and SOC. Observations are shown considering individual parent materials and horizons, corresponding to different respective colours and shapes. Where Ah are squares, Bw diamonds and Cw triangles. Diagonally, density plots show observation frequencies for respective predictor variables are displayed and split into individual parent materials, according to different allocated colours. In the right half of the matrix, spearman correlations coefficients for all soil mineralogy model predictor variables and SOC, considering all observations (Overall Corr) and individual parent materials, corresponding to different respective colours. Note that "\*\*\*", "\*\*", "\*", " " indicate p-values of < 0.001, < 0.01, < 0.05, < 0.1 and  $\geq$  0.1 that are allocated to the correlation coefficients. Dolomite observations are excluded as its observations were not considered in the SOC model predictions.

Author contributions. The conceptualization of the study for this paper was done by MG and SD and MG. Development of the methodology was done by SD and MG. Samples were collected and investigated by MM and MG. Data curation and validation was done by MM and AM. All formal analyses and visualizations were performed by AM except for the statistical model, which was performed by MM. AM wrote the final draft, based loosely off an earlier version by MM. All authors contributed to the editing and the revision of the paper.

Competing interests. The authors declare no competing interests.

Acknowledgements. The authors would like to thank Chigusa Keller, Loïc Imsand, Daniel Abgottspon, Anna Stegmann, Basil Frei and Julia Mayrock for assistance during sample collection. Furthermore, the authors would like to thank Julia Franzen for assisting in sample preparation and Jessica Carilli for conducting some of the laboratory analyses. The authors acknowledge the use of artificial intelligence (AI) in revising small segments of the R code for the data analysis pipeline.

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
