# Peer review of "Parent material geochemistry - and not plant biomass - as the key factor shaping soil organic carbon stocks in European alpine grasslands"

_EGUsphere, 2025_

## Author Comment (AC1)

**Author responses to comments for Maier et al. Biogeosciences preprint manuscript**

**Referee 1:**

**Overview Ref 1:** Maier et al studied the distribution and mechanisms controlling SOC distribution along a geochemical gradient of soil parent materials. The manuscript is overall well-written and the analyses are appropriated. The study compared plant inputs, soil characteristics and parent material in explaining the SOC stocks along the geochemical gradient and pointed out that "Parent material geochemistry – and not plant input - as the primary element shaping soil organic carbon stocks in European alpine grasslands". Please find below my major and minor concerns.

**Our response:** We appreciate the positive assessment and support from Reviewer #1 and their clear summary of the most important points in our manuscript. We also thank the reviewer for their comments that have helped us greatly improve the manuscript. We have provided a detailed point-by-point response below and highlight, text that we will add to the revised manuscript version in red.

**Ref1C1:** From the experimental results to European alpine grasslands. Because the experimental setting is along the geochemical gradients of soil parent materials, so the sites are selected to represent the difference in parent material. Comparisons among selected sites showed that parent material is important. Is it fair to conclude that parent material is more important than input for (entire) European alpine grasslands? Would the conclusion be different if you select the experiment sites across a plant input gradient?

**Our response:** Thank you very much for your valuable comments. Following up on the first part of the reviewer's comment, we do not yet have a definite answer to the generalizability of our results to the entirety of European alpine grasslands. However we will provide an answer to this question by performing a GIS-driven exercise for the revised version of the manuscript, identifying which grassland regions of the Alps fall within our geologic and climatic boundary conditions. This will entail the production of maps and the extraction and calculation of numerical values from the extent of the European Alps that show 1) the above treeline alpine grassland/meadow area underlaid by the major rock groups to which our soil parent materials belong to, 2) the area covered by the approximate ranges of our MAT and MAP ranges of our sampling sites i.e. approximately 2 ± 1 °C and 1210 ± 100 mm, respectively. The main information and insights gained from these analyses will be added into our discussion and conclusion sections in order to discuss how representative our results are regarding the entirety of European alpine grasslands extent. We will add the resulting coverage data into our supplement and potentially also add the resulting maps.

Furthermore, we will add the following text to our methods section in the revised manuscript describing how we will undertake these calculations and spatial

evaluations under subsection 2.3.8. that we will name 'Calculation and mapping of geologic and climatic boundary conditions'.:

To understand the generalizability of the results from this study to all European alpine grasslands, we analyzed the extent to which our geologic and climatic boundary conditions apply within QGis (3.40.5-Bratislava). This entailed the production of maps and the extraction and calculation of numerical values that show 1) the above treeline alpine grassland/meadow area underlaid by the major rock groups our study's soil parent materials belong to, 2) the area covered by the MAT and MAP ranges of our sampling sites, i.e. 2 ± 1 °C and 1210 ± 100 mm.

Alpine grassland/meadow data was extracted from Marsoner et al. (2023)'s detailed land use/landcover map for the areas included in the European Strategy for the Alpine Region, with a spatial resolution of up to 5 m and a temporal extent from 2015 to 2020. It was created by aggregating 15 high-resolution layers resulting in 65 land use/cover classes. The overall map accuracy was assessed at 88.8%. Herein, Alpine natural grassland was defined as being > 2000 m elevation. For the calculation of elevation, the European Digital Elevation Model (EU-DEM), version 1.1 was used (European Union, 2016). European geology coverage was taken from the European Geological Data Infrastructure (EGDI) 1:1'000'000 (OneGeology-Europe / EGDI, n.d.). Swiss geology coverage was extracted from BAFU 1:500'000 (Federal Office of Topography swisstopo, 2025). MAP data (30 arc sec, ~1km, based and a temporal extent from 1970-2000) was derived from WordClim Version 2.1 (Fick & Hijmans, 2017) and MAT data (30 arc sec, ~1km, and a temporal extent from 1981-2010) from Climatologies at high resolution for the earth's land surface areas (CHELSA) (Karger et al., 2017; Karger et al., 2018).

The previous section 2.3.8. 'Statistical analyses' will become the new subsection 2.3.9., with the same title.

In response to the second part of the comment, we would like to highlight that our analysis compared a range of soils developed on different parent materials, that also varied in maximum potential plant input (standing above- and belowground biomass stocks). The chosen locations are all located between 2000–2300 m.s.l., similar topographic position (slope and exposition, which will be added to Table 1) and cover a mean annual temperature range of 1.4–2.8 °C and a mean annual precipitation range of 1050–1300 mm. Our sampling sites cover alpine grasslands located on five geochemically distinct geologies of the European alps, according to geological maps (Federal Office of Topography swisstopo, 2020; Federal Institute of Geosciences and Natural Resources, 2020). The conclusion made in the paper, based on the selection of our five sites, that also varied in plant input, therefore remains the same, that parent material geochemistry was more important than plant biomass stocks in relation to SOC stocks.

**Newly added references:**

Marsoner, T., Simion, H., Giombini, V., Egarter, V.L., Candiago, S.: A detailed land use/land cover map for the European Alps macro region, Sci. Data, 10(1), 468, 10.1038/s41597-023-02344-3, 2023.

European Union, Copernicus Land Monitoring Service. European Digital Elevation Model (EU-DEM), Version 1.1. https://land.copernicus.eu/pan-european/satellite-derived-products/eu-dem/eu-dem-v1., 2016.

OneGeology-Europe / EGDI. (n.d.). Surface lithology of Europe (harmonized pan-European geology) [Web Map Service]. European Geological Data Infrastructure (EGDI). Retrieved August 5, 2025, from https://maps.europe-geology.eu/

Federal Office of Topography swisstopo: Lithological map of Switzerland 1:500000, https://map.geo.admin.ch, 2025.

Fick, S.E., Hijmans, R.J.: WorldClim 2: New 1-km spatial resolution climate surfaces for global land areas. International Journal of Climatology, 37(12), 4302–4315. https://doi.org/10.1002/joc.5086, 2017.

Karger, D.N., Conrad, O., Böhner, J., Kawohl, T., Kreft, H., Soria-Auza, R.W., Zimmermann, N.E., Linder, P., Kessler, M.: Climatologies at high resolution for the Earth land surface areas. Scientific Data. 4 170122. https://doi.org/10.1038/sdata.2017.122, 2017.

Karger D.N., Conrad, O., Böhner, J., Kawohl, T., Kreft, H., Soria-Auza, R.W., Zimmermann, N.E, Linder, H.P., Kessler, M.: Data from: Climatologies at high resolution for the earth's land surface areas. EnviDat. https://doi.org/10.16904/envidat.228.v2.1, 2018.

**Ref1C2:** Soil fertility vs. soil mineralogy model. The authors build random forest models with 42 samples and multiple explainable variables. Do these models face overfitting issues? What insights can we gain from these analyses, especially differentiating soil fertility vs. mineralogy? I feel the rationale is not very well justified. Why not have plant input as an explainable variable, and why not plant input, fertility and mineralogy together explain SOC variations?

**Our response**: Thank you for this critical feedback. The main incentive of our decision to design two separate prediction models for SOC with distinct sets of variables is to investigate whether SOC is more related to stabilization mechanisms (mineralogical variables) or to fertility parameters that may govern C input (fertility variables). Clay can be seen as both a proxy variable for C stabilization potential (Georgiou et al. 2022) and also as a proxy for the retention of water and nutrients in the soil, thus contributing to overall soil fertility (Kleber et al., 2015; Yu et al. 2022). Since clay content is a proxy for both and multifunctional in that sense, we decided to leave it in both models. If both predictor variable sets were merged into one prediction model, this would most

certainly result in overfitting, due to the very low degrees of freedom of the model (not enough observations). Although we already allude to clay as a proxy for the retention of nutrients (lines 240–242) in the methods of our manuscript, we do not explicitly address the reasoning behind the inclusion of clay in both of our models. We will therefore add the following statement after line 246:

"Clay is included in both models to act as a proxy for nutrient retention and water holding capacity of the soil (Yu et al. 2022), in the case of the soil fertility model, and as a proxy for SOC stabilization potential in the soil mineralogy model, as has been commonly done in SOC prediction models (Abramoff et al. 2021, Georgiou et al. 2022)."

Plant input in our study is not measured directly - we use total plant biomass and above-/belowground plant C stocks as proxy for potential biomass inputs. Furthermore, as is shown in Fig. 3 of the manuscript, the plant biomass C stocks do not consistently coincide with SOC stocks. We were mainly interested in soil-driven variables as predictors for SOC rather than trying to produce the best possible model for SOC (%) prediction.

We would nevertheless like to mention this in the manuscript explicitly in the methods section 2.3.7 'Calculation of plant biomass and soil organic carbon stocks' after line 212:

'Plant biomass OC stocks will be interpreted as a proxy for potential plant biomass C input, as plant input rates were not measured directly within this study'.

In regards to the overfitting question, we re-ran our random forest models to account for spatial autocollinearity by using an adapted leave-one-plot-out cross validation (CV). The leave-one-plot-out CV measure is used to reduce overfitting in regression statistics with limited field observations (Yates et al. 2022). We present the results of these revised models as a revised version of Figure 5., which we will add to the revised manuscript. This revised Figure has the same structure as the current Fig 5. in the manuscript and shows the predicted bulk SOC (%) vs. measured/observed bulk SOC (%). Accordingly, we will change the individual reported numerical values such as the $R^2$, *RMSE* and the relative importance values of bio-P and clay denoted in the results section 3.4. so that they align with the newly calculated values. Due to the similar results from these revised models, we would leave the discussion section 4.4. as is. We will add the change lines 253–254 of our current methods section 2.3.8. to following to describe the adapted cross validation:

We used an adapted leave-one-out cross validation (i.e., leave-one-plot-out CV) to account for spatial autocollinearity within plots and thus to reduce overfitting (Yates et al. 2022). Iterating through all plots, only one plot and its observations were taken as test data, while the rest of the dataset was used as a training set, until all plots were used once as test data, on which we tested the prediction accuracy of our model.

[Figure]

**Revised Figure 5. (a)** Predicted bulk SOC (%) values vs. measured/observed bulk SOC (%) values of the soil fertility model. **(b)** Predicted bulk SOC (%) values vs. measured/observed bulk SOC (%) values of the soil mineralogy model. **(c)** Relative importance (%) of the model predictor variables included in the soil mineralogy model. **(d)** Relative importance (%) of the model predictor variables included in the soil fertility model. The dashed line represents a 1:1 regression line.

In comparison to the model structure of that presented in the current Fig 5. in the manuscript, the *RMSE* of both the soil fertility and mineralogy model increased slightly (from 2.02 to 2.14 and from 2.53 to 2.62 respectively) and the $R^2$ of both models decreased slightly (from 0.91 to 0.86 and from 0.84 to 0.81 for the soil fertility and soil mineralogy model respectively). The most important predictor variables remain the same as those run with the repeated 10-fold cross validation method for both models. However, their relative importance to the models increases from a previous 20.1% to 32% for bio-P for the soil fertility model and from 24.2% to 44.5% for clay in the soil mineralogy model. The sequence of the other individual predictor variables' relative importance remains very similar for the soil mineralogy model, with only Fe$_{PP}$ becoming seemingly more important than Al$_{PP}$. The sequence of the other individual predictor variables' relative importance varies more for the soil fertility model, yet the magnitude of importance of the individual predictors remains very similar.

**Newly added references:**

Abramoff, R.Z., Georgiou, K., Guenet, B.: et al.: How Much Carbon Can Be Added to Soil by Sorption?, Biogeochemistry, 152(2), 127–42, https://doi.org/10.1007/s10533-021-00759-x, 2021.

Georgiou, K., Jackson, R.B., Vindušková, O., et al.: Global Stocks and Capacity of Mineral-Associated Soil Organic Carbon, Nature Communications, 13(1), 1, https://doi.org/10.1038/s41467-022-31540-9, 2022.

Yates, L.A., Aandahl, Z., Richards, S.A., Brook., B.W.: "Cross Validation for Model Selection: A Review with Examples from Ecology", Ecological Monographs, 93(1), e1557, https://doi.org/10.1002/ecm.1557, 2023.

Yu, M., Tariq, S.M., Yang, H.: Engineering clay minerals to manage the functions of soils, Clay Minerals, 57, 51–69, https://doi.org/10.1180/clm.2022.19, 2022.

**Ref1C3:** Soil aggregates. The study found that microaggregate contributes to a large (>50%) portion of bulk SOC, and inferred those sites with metal oxides favored the formation of microaggregate. Are there approaches to provide more direct evidence from this experiment?

**Our Response:** Thank you for raising this question. We will provide a new figure (see Figure S2 below) to the supplement, that shows the relationship between **(a)** Total Fe concentrations in the bulk soil (g kg$^{-1}$) vs. the the microaggregates (MA) weight % (wt%) to the bulk soil, **(b)** the sum of iron concentrations from both the pyrophosphate-extractable Fe ($Fe_{PP}$) and ammonium-oxalate extractable Fe ($Fe_{AO}$) (i.e. the sum of the reactive metal oxides) vs. the the microaggregates (MA) weight % (wt%) to the bulk soil, **(c)** the sum of iron concentrations from the pyrophosphate-extractable Fe ($Fe_{PP}$), ammonium oxalate-extractable Fe ($Fe_{AO}$), and dithionite-citrate-bicarbonate-extractable Fe ($Fe_{DCB}$) (i.e. all extractable metal oxides- reactive + more crystalline oxy(hydr-)oxides) the microaggregates (MA) weight % (wt%) to the bulk soil.

[Figure]

**Figure S2. (a)** Total Fe concentrations (g kg$^{-1}$) vs. MA (wt%), **(b)** sum of Fe concentrations from both Fe$_{PP}$ and Fe$_{AO}$ vs. MA (wt%), **(c)** sum of Fe concentrations Fe$_{PP}$, Fe$_{AO}$, and Fe$_{DCB}$ vs. MA (wt%) . Observations from the Dolomite site were excluded from this figure as its Oh horizon is organic and showed signs of POM contamination in the MA fraction. Note that the individual panels show different y-axis ranges but share the same x-axis range.

In Figure S2, the role of Fe is shown to be connected to the mass of microaggregates (MA) in the soils. This trend can be seen for different iron phases including the total Fe concentrations within the bulk soil ((a)) as well as the reactive (Fe$_{PP+AO}$) ((b)) and total Fe pedogenic oxide concentrations (Fe$_{PP+AO+DCB}$) ((c)). While the correlation between Fe$_{tot}$ and MA mass is not significant (Spearman correlation coefficient of 0.16 with $p > 0.1$), those between Fe$_{PP+AO}$ and MA mass and Fe$_{PP+AO+DCB}$ and MA mass are positive and significant (Spearman correlation coefficients of 0.53 and 0.48 respectively, with $p < 0.05$). We acknowledge that we cannot derive causation based on correlations, however, the relationship of Fe oxides with microaggregates (Fig. S2)

aligns with literature, which explored the importance of reactive metals for aggregation processes (Lehndorff et al., 2021, Campo et al., 2024, Totsche et al. 2018). On the other hand, total and pedogenic oxide Al concentrations do not correlate as well with the MA mass. Further, the concentrations of total extractable pedogenic oxides of Fe compared to Al are on average 2.5 ± 1.1 (SD) times larger. We interpret this as a consequence of the pH range in most of the sampled soils not being low enough to lead to heightened Al mobility. Mn concentrations may have contributed to aggregation processes as well, however its concentrations are much lower than those of Fe (between 12-150 times smaller) and Al (between 11-99 times smaller) and hence its contributions are negligible (data not shown). Based on these findings we assume that Fe may play a more important role for aggregation processes in our examined soils. We would like to add a small statement to the discussion summarizing the correlations from the supplementary Fig. S2 together with a more in depth discussion of literature findings that support our interpretation of a significant role of microaggregates for C stabilization. We also want to discuss the connection of these fractions to pedogenic oxide concentrations. We suggest to slightly change the phrasing discussion of the lines 442–446 and augment the discussion at this location of the manuscript with the discussion of additional literature as follows:

'Wasner et al. (2024) examined a geoclimatic gradient across a diverse range of grassland topsoils and found that the stable microaggregate fraction was the biggest contributor to bulk SOC in C-rich soils. They report that both the SOC quantity in free silt and clay (s+c) and stable microaggregates (MA) fractions were positively correlated to pedogenic oxide contents and texture. The results from this publication support the notion that stable microaggregates can be major contributors to bulk SOC in grassland soils across large environmental gradients. Another publication by Lehndorff et al., (2021) examined the spatial organization of soil microaggregates in a sandy and a loamy Luvisol. They report greater OC concentrations within the microaggregate fractions compared to bulk SOC, and their analyses support the notion that OC forms the core for microaggregate formation and is protected within microaggregate structures. They also found a systemic increase in iron, followed by clay and silicate mineral phases in the microaggregates formed on clay-richer soil, indicating that the inherent soil mineralogy is reflected in the composition of microaggregates. Their data further supports the notion that pedogenic iron aids aggregation processes in the soil by acting as a cementing agent (Campo et al., 2024). The importance and prevalence of iron in microaggregates is therefore shown to be a supporting phase for overall microaggregate stability, and thus ultimately supports the retention of SOC therein. This high relevance of Fe in particular for aggregation processes is also reflected in significant correlations of reactive Fe pedogenic oxides ($Fe_{PP+AO}$) and total Fe pedogenic oxides ($Fe_{PP+AO+DCB}$) with the MA mass (wt%) for our sites (Spearman correlation coefficients of 0.53 and 0.48 respectively, with $p < 0.05$, Fig. S2). While no significant correlations could be found for reactive or total Al or Mn pedogenic oxides (data not shown). Due to an overall significantly lower amount of total Mn and extractable Mn pedogenic oxide concentrations, we assume the

contribution thereof to be rather little to aggregation processes (data not shown). We interpret the insignificant connection of Al pedogenic oxides to MA mass to be related to the pH range of most of our sites, which is above values where Al becomes significantly mobilized and ranges generally between 4.2–6.8, except for the Granite site (3.8 ± 0.2 (SD)).

We would also like to slightly adapt and augment the summary paragraph beginning from line 446 onward:

'[...] a large contribution of MA was found to contribute to bulk SOC, that appear to be supported by the presence of reactive, pedogenic Fe phases. These findings underline the importance of aggregation processes for SOC stabilization in alpine soils developed on different parent materials.'

Lastly, with the addition of these additional correlation analyses we would like to add a brief description thereof in the methods section 2.3.8. from line 232 onwards: 'Spearman correlations were used to examine relationships between select soil biogeochemical variables, as this non-parametric method is appropriate for data that are non-normally distributed.'

**Newly added references:**

Campo, J., gimeno-Garcia, E., Andreu, V., Gonzalez-elayo, O., Rubio, J.L.: Cementing agents involved in the macro- and microaggregation of a Mediterranean shrubland soil under laboratory heating, Catena, 113, 165-176, https://doi.org/10.1016/j.catena.2013.10.002, 2014.

Lehndorff, E., Rodionov, A., Plümer, L., Rottmann, P., Spiering, B., Dultz, S., Amelung, W.: spatial organization of soil microaggregates, Geoderma 386, 114915, https://doi.org/10.1016/j.geoderma.2020.114915, 2021.

Totsche, K. U., Amelung, W., Gerzabek, M.H., et al.: Microaggregates in Soils, J. Plant Nutr. Soil Sci., 33, 104-136, https://doi.org/10.1002/jpln.201600451, 2018.

**Ref1C4:** Please double check on figures. There seem swaps. For example, Figure 3a, Gneiss, the crosshatch and solid part seem did not follow the texts. The crosshatch is much big, but in the text, it states that the wood (solid) is bigger. Figure 5 c, should it be the soil fertility model, but in the captions, it states "soil mineralogy model". This swap also affects Figure 5d

**Our Response:** Thank you for this comment. We will rewrite the Fig. 3 caption to more clearly delineate that the crosshatch pattern represents the woody biomass contribution and that the solid part only represents the herbaceous/non-woody biomass contribution. In the text we already explicitly mention the 80% contribution of woody biomass to total whole profile biomass for the Gneiss site, but we will add a small sentence saying that for all other sites the woody biomass contribution to aboveground biomass C stocks is either zero or < 20 %. We thank you for pointing out the swaps in Figure 5 and will adjust them accordingly.

We will adjust Figure 3 (a) caption as follows: 'The relative amount of woody aboveground plant biomass is represented by the black crosshatch pattern within the bars, and the solid part of the bars represents the relative amount of herbaceous/non woody biomass [...]'

We will add the following information to line 294: 'Of which an approximate 80 % consists of woody plant material. For all other sites the woody biomass presence is either zero or < 20 %.

---

## Author Comment (AC2)

**Author responses to comments for Maier et al. Biogeosciences preprint manuscript**

**Referee 2:**

**Overview Ref 2:** Maier et al., 2025 compare observations from European alpine grassland soils developed in five geochemically-distinct parent materials to assess how plant biomass, soil fertility, or geochemical variables are related to SOC stocks. The authors then use physical fractionation and $^{14}$C analyses to evaluate the distribution of carbon and its persistence (in the bulk soil). The study reconfirms a well-established correlation between Fe, Al, and Mn concentrations, extracted with classical-sequential extractions, and SOC stocks, while also reporting a large contribution of the microaggregate fraction to bulk SOC. Overall, I enjoyed reviewing this article and found that their data was well presented, and the study, well referenced. I thought that the study benefited from a well-thought out sampling strategy, which took into account landscape heterogeneity within their sites, and the difficulties of getting representative samples from grasslands in environments with complex topography.

My comments regarding the manuscript are largely minor, but are numerous. Instead of separating my comments into major or minor revisions, I've instead presented the comments as a line-by-line evaluation, with more general comments at the beginning of each section. My main two general comments are in connection to the language and figures. Please carefully evaluate your terminology throughout and be more specific with its usage in the manuscript. I've also made some suggestions on your existing figures and would ask for the inclusion of more supplementary figures to support your narrative and discussion. With these corrections, this paper will shortly be ready for publication in Biogeosciences.

**Our response:** We appreciate the kind words and positive support from Reviewer #2 and their detailed reading of our manuscript. We thank the reviewer for their extensive comments that helped us improve the manuscript significantly. We have provided a detailed point-by-point response below and highlight, text that we will add to the revised manuscript version in red.

**Title:**

**Ref2C1:** Is element the right word here? Would driver, (controlling) factor, or variable be better suited?

**Our response:** Thank you for this comment. We see that the word 'element' is not ideal, hence we suggest using the following, alternative title:

Parent material geochemistry – and not plant biomass – as the key factor shaping soil organic carbon stocks in European alpine grasslands

**Abstract:**

**Ref2C2:** Line 4: Is distribution the right word here? Or should it instead be SOC stocks? And is it SOC stabilisation mechanisms or instead geochemical variables?

**Our response:** Thank you for your questions regarding our phrasing and your suggestions.

Line 4: We will change the word 'distribution' to 'SOC stocks'. In line 7 we do mean stabilization mechanisms with regards to the differing contributions of the examined soil fractions to bulk SOC, but we do also show that geochemical variables such as pedogenic oxide content matter for SOC stabilization. So we will additionally mention geochemical variables here as well:

we suggest changing line 6 to: '[...] in the importance of geochemical variables and SOC stabilization mechanisms [...]'

**Ref2C3:** Line 10: At the moment, this reads that soil fertility governs plant C inputs and soil mineralogical characteristics. I'd suggest instead combining "soil fertility and soil mineralogical characteristics, which govern plant C inputs and control C stabilisation respectively, play equally critical roles"….

**Our response:** We will change the sentence to your suggestion to make it more clear: 'Our results highlight that soil fertility and soil mineralogical characteristics, which govern plant C inputs and control C stabilization respectively, play equally crucial roles in predicting SOC contents in alpine soils at an early development stage.'

**Ref2C4:** Line 11: Could the F14C values be mentioned earlier? As the previous sentence quite nicely summaries the paper.

**Our response:** Thank you for this comment, we agree that this makes sense. And suggest changing the sentence beginning in line 4 to the following: 'We demonstrate that SOC stock accrual and persistence in geochemically young soils, with fraction modern ($F^{14}C$) values ranging from 0.77–1.06, is heavily dependent on soil mineralogy as a result of parent material weathering, but is not strongly linked to plant biomass.'

**Introduction:**

**Ref2C5:** Line 19: Is this in reference to a specific study? It seems so. If so, please add the citation.

**Our response:** Thank you for this question. This is a reference to the study from Rumpf et al. (2022), which we cite in Line 20. However, we will add an additional in-text citation to this study in line 19.

**Ref2C6:** Line 24: This sentence has an abrupt break after understudied, "remain understudied" would probably be better suited at the end of the sentence.

**Our response:** Thank you for pointing this out, we will add "remain understudied" at the end of the sentence.

**Ref2C7:** Line 32: and short-range order minerals / oxy(hydro)oxides?

**Our response:** Yes thank you for pointing this out, we will mention their importance at the end of the sentence as well:

'Weathering also leads to increased formation of reactive secondary minerals, such as expandable clays and short-range order minerals (SRO)/oxy(hydr-)oxides.'

**Ref2C8:** Line 40: As a suggestion it could be nice to clarify and conclude why this paragraph has been important to your narrative in your own words. It ends rather abruptly. This occurs throughout the manuscript.

**Our response:** Thank you for this comment and your attentive reading. We will add the following paragraph from line 40 onwards to conclude the importance of the knowledge from this paragraph:

'Many of the above-described SOC stabilization mechanisms are assumed to be underdeveloped in alpine soils that are geochemically younger, less altered, and have not undergone intense weathering in the past. This is linked to the climate conditions found in alpine ecosystems which typically lead to a slower - or limited- formation of alpine soils compared to lowland soils (Körner, 2003; Doetterl et al., 2018). However, few studies have examined the relative importance of plant biomass-derived C inputs into the soil compared to SOC stabilization potentials of alpine soils for SOC stock accrual.'

In the discussion section of the manuscript we tried to conclude the importance gained from the paragraphs in a small summary paragraphs and we believe we already adequately convey why the information discussed is relevant to our own narrative here in most subsections. We however would like to add the following brief statement following section 4.4., after line 468:

'In summary, soil fertility as well as soil mineralogical parameters performed equally well for predicting SOC content and can therefore ultimately both contribute to SOC accrual in these shallow to moderately deep, developing alpine soils.'

In some other sections of the manuscript we have now amended some more text e.g. from lines 445/446 onwards (see response to **Ref1C3 and below Ref2C49**) and have further summarized that newly described knowledge in the summary statement following the main discussion of that respective subsection. With these new amendments, we hope to have sufficiently addressed your comment and will keep this in mind when submitting the revised version of the manuscript.

**Ref2C9:** Line 41-49: Climate change could also enhance the decomposition of C at elevation and it would be good to present this critical counterpoint. By increasing the temperature of soils, increasing the efficiency of decomposers, their extra-cellular enzymes, changing moisture regimes, and increasing the mineralisation / respiration of SOC. It would be good to provide both sides of the argument here. I believe that Garcia Franco et al., (2024; Geoderma 442, 116807) may be a good reference in this case.

**Our response:** Thank you for this critical feedback. This is correct, and we will definitely mention both perspectives of the argument. We suggest adding the following sentences before the existing paragraph that currently starts at line 41:

'The lack of interactions that enable SOC stabilization in alpine soils has been shown to result in a heightened vulnerability of OC stocks to changes in climate (Parker et al., 2015; Hagedorn et al., 2019; Walker et al., 2022; Garcia-Franco et al., 2024). However, climate warming will also induce an acceleration in soil weathering rates and improve climatic growth conditions of high-altitude vegetation, which may also lead to the development of larger SOC stocks.'

**Ref2C10:** Line 51: I'd suggest a paragraph break here to separate your study design from the knowledge gap and then combine the latter part of this paragraph with the following paragraph containing your hypotheses.

**Our response:** Thank you for this suggestion, we will relocate the current small section from line 51-56 to after the paragraph describing our research hypotheses.

**Ref2C11:** Line 60: I'd also suggest being more succinct in your hypotheses in this final paragraph of the introduction.

**Our response:** Thank you for this comment. We suggest to shorten the hypotheses to the following version:

'First, we hypothesize that i) soils developed on nutrient-rich parent materials will exhibit the highest biomass inputs and potentially also the highest overall SOC stocks, if soils can stabilize these inputs (e.g., through high clay content or pedogenic oxides). More specifically, we expect that ii) soils with a higher amount of pedogenic oxides– and not the amount of clay as a particle size–will lead to greater SOC stocks, because of the enhanced stabilization potential of those minerals (Rasmussen et al., 2018). Lastly, we expect iii) soils that are more strongly weathered will not only show higher SOC stocks but also older $^{14}$C values, hinting at higher persistence, by having developed a wider portfolio of efficient mineral-related stabilization mechanisms, resulting in better protection of SOC against microbial decomposition (Kleber et al., 2015; Torn et al., 2013).'

**Ref2C12:** Line 70: conjoined parentheses - ) ( could be combined with a semicolon throughout. (POM; Kleber et al., 2015; Torn et al., 2013). If using Endnote, this can be achieved with the prefix function. Please change this throughout.

**Our response:** Thank you for this comment. We will change this as suggested throughout the document.

**Materials and Methods:**

**Ref2C13:** Line 82: Nested parentheses - Brackets in brackets are typically square brackets or a different format. Please change this throughout.

**Our response:** Thank you for this comment. We will change this as suggested throughout the document.

**Ref2C14:** Line 89: This is an open question, but does this study assume similar age of each soil type since deglaciation? How could slight variations in your soil forming factors have influenced your interpretations? Is it really geochemistry that is defining your SOC stocks or just time and the expression of pedogenesis?

AC: Thank you for this question. We did not measure the age of the soils themselves, only their bulk $F^{14}C$ values but are confident that we can assume a similar development age for each soil type since deglaciation as we chose our sites carefully to resemble similar climatic, orographic, and topographic conditions. Thus, we held the soil forming factors *climate*, *topography*, and *time* as constant as possible across the sites. We acknowledge that some variation in deglaciation may be present across are soils, which would change the time from when weathering commences. Yet even if there were different amounts of years since deglaciation across the sites, studies have shown that organic matter accumulation often shows an initial period of rapid increase for up to 3,000 years, followed by a lower accumulation rate that may continue for millenia (Birkeland, 1984). We therefore assume that 10,000 years post deglaciation the soils we examined are in a state where they have reached their long-term SOC accumulation rates (Schlesinger, 1990). Furthermore, we could not identify any recent disturbances such as landslides that would infer major interferences with soil development. There is certainly variation in pedogenesis across our sites, especially given the differences in parent material. In fact, this is a central aspect to our study and in our study design, i.e., allowing us to study changes in C dynamics as a result of geological/geochemical differences and the corresponding responses in vegetation and soil.

**Ref2C15:** I see that aspect and slope angle was kept as constant as possible with your study design, but could you please add it to Table 1 if you have it?

**Our response:** Yes, we will add slope and aspect to Table 1.

**Ref2C16:** Furthermore, I will trust your classification, but is this definitely a ~15 cm thick Oh horizon atop a R horizon? There is no Ah horizon here?

**Our response:** Yes, there was no Ah horizon present (please see picture below). It was only an Oh horizon directly on top of an R horizon, a typical feature of soils developed on dolomite and other Ca-rich rocks at this elevation (e.g. D'Amico et al, 2023, IUSS Working Group WRB, 2022).

[Figure]

**Photo**: Example of a soil profile at the dolomite site with the Oh horizon (indicated by its dark colour and SOC concentrations >20%) lying directly on top of the parent material.

**References:**

D'Amico, M.E., Casat, E., Abu El Khair, D., Cavallo, A., Barcella, M., Previtali, F.: Aeolian inputs and dolostone dissolution involved in soil formation in Alpine karst landscapes (Corna Bianca, Italian Alps), CATENA, 230, 107254, 10.1016/j.catena.2023.107254, 2023.

IUSS Working Group WRB. World Reference Base for Soil Resources. International soil classification system for naming soils and creating legends for soil maps. 4th edition. International Union of Soil Sciences (IUSS), Vienna, Austria., 2022.

**Ref2C17:** Principal qualifiers are capitalised.

**Our response:** Thank you for this comment. We will capitalize all qualifiers in the next version.

**Ref2C:** Line 90: Calcareous?

**Our response:** Yes, that is the correct word that we were looking for. We will replace the former word with this one.

**Ref2C18:** Line 94: If I'm not mistaken, Calcaric is the principal qualifier in connection with Cambisols, again capitalised. Calcic would make your profile a Calcisol (Calcic horizon) and is not typically applied to a Cambisol.

**Our response:** Thank you for pointing this out. We agree that calcaric is not a principal qualifier connected with Cambisols, however according to WRB (2015) if a qualifier applies but is not in the list for a specific reference soil group, it may be added as a supplementary qualifier in brackets after the name of the reference soil group. Hence we suggest renaming our previous 'Calcaric Cambisol' to a 'Cambisol, (calcaric)'.

**Ref2C19:** Line 99: September,

**Our response:**Thank you, we will add the comma.

**Ref2C20:** Line 101: You can remove the % after 40 as it is repeated after 60.

**Our response:** Thank you, we will remove the superfluous %.

**Ref2C21:** Trough is a strange old English word in this context, meaning a long narrow open container for animals to consume from, or a channel for moving water. I don't think that you referring to tillage (trough) here? Is it instead micro-elevational changes at your grassland? Please clarify.

**Our response:** Thank you for catching this error. We translated a specific German word but must have missed the correct meaning in English. What we meant were small areas that are slightly lower than others leading to water accumulation (and thus to altered SOC dynamics). The correct word might be 'depression', which we will use to replace 'trough'.

**Ref2C22:** Line 117: Ground would also be applicable or just milled.

**Our response:** We will remove the 'powder' part before milled.

**Ref2C23:** Line 126-128: I might have missed this, but I didn't see mention of this in 2.3.8. What was the effect of their inclusion?

**Our response:** Thank you for this question. We meant to say that the same Marl plots that are excluded from the physical fractionation were also excluded from the statistical analyses. We will adapt the text description so that it is more clear as follows:

We would suggest removing the mention of 'These correspond to the same plots that were included for statistical analyses (see details in Sect. 2.3.8).' as this text may be confusing.

However, we suggest adding the following statement after line 226 in section 2.3.8.: Please note that the respective sampling plots of Marl, that displayed a parent material geochemistry that did not correspond with the geochemical composition of a typical Marl, exhibiting markedly higher Si contents, were excluded from statistical analyses.'

**Ref2C24:** Line 142: Have you got a ref for the pH measurement? It's normally achieved with different timing and soil-to-solution ratios so it would be good to include the ref.

**Our response:** Thank you for pointing this out. We applied a protocol similar to that of Rayment & Lyons (2011) but with slight modifications. We will cite them and mention this in our manuscript. In contrast to their method where they shake the soil and $CaCl_2$ solution samples for 1 h overhead, we shook the soil and solution at max speed on a horizontal shaker for 10 minutes and measured samples 24 h later, instead of within 4 hours of shaking. Please note that in our research group we have measured pH with this method at different time points, following the settling of particles post shaking, and the pH value stayed stable over the 24 h time period.

**Ref2C25:** Line 143: Why were Si, Ti, and Zr measured with XRF and the rest of the elements measured with a pseudo-total digest using aqua regia? Didn't you get these elements from the XRF measurements?

**Our response:** Thank you for this question. Si does not properly digest with aqua regia, leading to erroneous concentration values upon measurement with ICP-OES. In addition, Ti and Zr constitute high mass elements and Krishna et al. (2008) found

that the detection limit for such elements is generally better with XRF than ICP-OES. Although many more elements than these three can be detected with XRF, the same study finds that elements lighter than F are better detected with ICP-OES, as well as Na, Mg and Al. In addition, our sequential pedogenic oxide extractions were also measured on ICP-OES. Therefore we thought it best that the same sample preparation and measurement method be applied for the quantification of Fe, Al, Na etc. in order to have comparable values and to also produce ratios (e.g. $Fe_{DCB}/Fe_{tot}$).

**Ref2C26:** Line 154: It seems like only the Cw horizon of the Marl soil had a pH higher than 6. Did you also treat the Oh horizon of the Dolomitic Leptosol? Did you measure the stable isotopic C composition also? Your work assumes that carbonate removal was complete, but was it?

**Our response:** Yes, we treated all samples with a pH > 6 (Cw of Marl and Flysch), since the Oh horizon of the Dolomitic Leptosol had a pH between 4.7 - 5.7 we did not treat these samples. We did not measure stable isotopic C composition. And yes, we do assume the carbonate removal was complete as we used a specialized, adapted method to specifically account for this. We would like to remove the current text from lines 152–155 and replace it with the following, more descriptive text to more accurately describe the methodology applied:

"Soils with a pH > 6 were treated before the analysis to remove the carbonates. The carbonate removal was conducted according to Ramnarine et al. (2011) using hydrochloric acid fumigation and adopted similarly as Peixoto et al. (2020). A beaker of 50 ml of 37% HCl was prepared and placed in a desiccator. The soil samples were added for a duration of 72 hours. Afterwards the beaker filled with HCl was replaced with NaOH pellets for the removal of residual HCl vapor for additional 72 hours. Each sample was measured without HCl treatment for accurate nitrogen contents (Walthert et al. 2010). "

**Ref2C27:** Line 176: This first sentence needs to be corrected. "To identify reactive metal phases that can interact soil C…" Be careful with the differentiation of correlation and causation throughout, particularly when referring to C stability.

**Our response:** Thank you for pointing this out. We will correct the sentence to following: 'To identify reactive metal phases that can interact with (and potentially stabilize) soil C, Fe, Al and Mn oxides were extracted sequentially using sodium pyrophosphate (PP) (Bascomb, 1968), ammonium oxalate (AO) (Dahlgren, 1994), and dithionitecitrate-bicarbonate (DCB) solutions (Mehra and Jackson, 1958).'

**Ref2C28:** Line 180: I appreciated your disclaimer here regarding the usage of sequential extraction and the table below. Even though these are well established extractions, it would however have been good to include how they were extracted, similar to your description of the pseudo-total digests. The CEC extraction could also be more detailed. 1 g of soil was extracted in 100 mL…

**Our response:** Thank you for these comments.

We will augment the sequential pedogenic oxide extraction methodology description from line 181 with the following:

"In brief, 0.5 g of 30 °C dried and milled bulk soil was extracted with a sodium-pyrophosphate (PP) solution at pH 10 (consisting of 0.1 M $Na_4P_2O_7$ x 10 $H_2O$ and 0.5 M $Na_2SO_4$) at a soil to solution ratio of 1:40 for 16 h on a horizontal shaker. Then the vials were centrifuged (Sigma 3–16 KL, 20 °C, 1,700 × g) for 10 min. The supernatant was decanted, filtered (Whatman 41) and diluted to 50 ml with MilliQ. These extracts were stored at 4° C prior to measurement on ICP-OES. The remaining soil residue was extracted with a 0.2 M ammonium oxalate (AO) solution (consisting of 0.2 M ammonium oxalate and 0.2 M oxalic acid dihydrate in a ratio of 1:31:1) at pH 3 and a soil to solution ratio of 1:40 for 2 h on a horizontal shaker in the dark to prevent photodegradation of the extract. Then the vials were centrifuged (Sigma 3–16 KL, 20 °C, 1,700 × g) for 10 min. The supernatant was decanted, filtered (Whatman 41) and diluted to 50 ml with MilliQ. These extracts were stored at 4° C prior to measurement on ICP-OES.

The remaining soil residue was placed in a pre-heated water bath (70–75 °C) with 0.3 M sodium citrate and 1 M $NaHCO_3$, at a soil to solution ratio of 1:22.5. To this 0.25 g sodium dithionite was added stepwise every 15 min until the solution was grey-coloured. After, 5–10 min of cooling outside the bath Then the vials were centrifuged (Sigma 3–16 KL, 20 °C, 4000 rpm) for 10 min. The supernatant was decanted, filtered (Whatman 41) and transferred into a clean vial. 0.5 ml of MilliQ were added to the leftover soil pellets and the vials were centrifuged again (Sigma 3–16 KL, 20 °C, 3000 rpm) for 10 min. The supernatants were transferred analogously into the vial containing the previous supernatant. This procedure was repeated once more. Then, 2.5 ml of magnesium sulfate was added to the soil pellets. Samples were vortexed, ultrasonicated and centrifuged again (Sigma 3–16 KL, 20 °C, 3000 rpm) for 10 min. The supernatants were transferred analogously into the vial containing the previous supernatants. These extracts were stored at 4° C prior to measurement on ICP-OES. "

And leave the text after the lines 180/181–186 as they are. They will follow the inserted text segment above.

We will augment the effective CEC methodology description following line 146 with the following:

"Effective cation exchange capacity ($CEC_{eff}$) and the amount of exchangeable cations (Ca, Mg, K, Na, Al, Fe, Mn) were determined according to Hendershot (1986) using $BaCl_2$. In brief, 20 ml of 0.1M BaCl2 were added to 3 g of soil in a 50 ml centrifuge tube. These samples were then placed on a horizontal shaker for 2 h at a setting of 150 rpm, centrifuged at 2500 rpm for 10 min and filtered (42 µm) into 50 ml tubes. Extracts were then stored at room temperature before measurement. Shortly before measurement on the Inductively Coupled Plasma Optical Emission Spectroscopy

(ICP-OES) (5100 ICP-OES, Agilent Technologies, Santa Clara, CA, United States), samples were diluted 1:10 with MQ and 0.1 ml of internal standard (in $HNO_3$, 65 %)."

**Ref2C29:** Were the ICP-OES measurements made with an internal standard? What was the relative standard deviation? Did you measure duplicates? How reproducible were your observations?

**Our response:** Yes, we used Scandium (Sc) measured at 361.383 nm as an internal standard. We measured all (n = 4) field replicates and included select laboratory replicates for a limited number of sites and specific horizons. For the ICP-OES measurements, the laboratory precision was within ± 2.2% (RSD), with a range of 1.7-2.9% (RSD) across the measurements of pedogenic oxides.

**Ref2C30:** Line 187: I'd suggest splitting the calculation of the fertility indices section from the F14C measurements as they're largely unrelated.

**Our response:** Thank you for this comment. We will separate the two into two distinct subsubsections leaving the determination of $F^{14}C$ in subsubsection 2.3.6 and placing the weathering indices calculation into the 'new' subsubsection 2.3.7.

**Ref2C31:** Line 201: *ca.* is typically italicised like *vs.*, throughout.

**Our response:** Thank you for this comment. We can write these abbreviated words in italic in the updated version.

**Ref2C32:** Line 206: Soil persistence?

**Our response:** Thank you for your comment. In our manuscript we are interested in SOC persistence, in addition to being interested in the abundance (concentrations, stocks, contents etc.) and incorrectly wrote 'soil' persistence. Thank you for pointing this out, we will change this accordingly in the manuscript.

Section 2.3.8:

**Ref2C33:** How did you test for normality?

**Our response:** Thank you for this question. We will add this as a statement in line 229: 'We tested for normality using visual inspection such as QQ and scale-location plots.'

**Ref2C34:** Did you evaluate changes with depth also? And if so, how do your statistics account for the spatial autocorrelation with depth? Could linear mixed models have been used to evaluate changes with depth?

**Our response:** Thank you for this question. Our study was mostly interested in changes of stocks across geologies at large and less with changes with depth within an individual site or across different sites, thus these were not evaluated extensively. Furthermore, our sampling design was set up so that our soils were sampled by pedogenic horizons, rather than within small depth increments. In that sense, our sampled soil horizons may not align with the same soil depths, across sites. We

wanted to sample horizon-explicitly as we were mainly interested in capturing the functional and developmental variability of the different horizons as the result of longer-term pedogenetic processes across our sites**.**

We would like to communicate our motivation for this sampling scheme in a bit more detail in the methods section 2.2. to line 105:

'In that sense, our sampled soil horizons may not align with the same soil depths, across sites. We wanted to sample horizon-explicitly as we were mainly interested in capturing the functional and developmental variability of the different horizons as the result of longer-term pedogenetic processes across our sites**.'**

We added Table 4. in the manuscript to showcase some of the depth trends within and across sites for selected bulk soil variables such as C:N (-), bioavailable phosphorous, total reserve in base cations, effective cation exchange capacity, the sum of Fe, Al and Mn pyrophosphate- and ammonium oxalate-extractable pedogenic oxides, Sand and Clay content and we will augment this table with the coarse fraction content as well for the revision.

To account for spatial autocorrelation within a plot (i.e., across depth as well), we reran our models by using an adapted leave-one-plot-out cross validation. In response to the next comment (see **Ref2C35**) we conducted a principal component analysis and produced a biplot that we will add to our supplement. In the next comment we also describe how bulk soil variables, including SOC, are related to one another and to soil 'Depth'. A variable that corresponds to the lower depth of a respective horizon, measured from the soil surface.

Furthermore, we received a similar comment from reviewer 1, where they inquired about potential overfitting issues. We responded the following (abbreviated here): We re-ran our random forest models to account for spatial autocollinearity by using an adapted leave-one-plot-out cross validation (CV). The leave-one-plot-out cross validation measure is used to reduce overfitting in regression statistics with limited field observations (Yates et al. 2022). We present the results of these revised models as a revised version of Figure 5., which we will add to the revised manuscript. This revised Figure has the same structure as the current Fig 5. in the manuscript and shows the predicted bulk SOC (%) vs. measured/observed bulk SOC (%). Accordingly, we will change the individual reported numerical values such as the $R^2$, RMSE and the relative importance values of bio-P, Clay denoted in the results section 3.4. so that they align with the newly calculated values. Due to the similar results from these revised models, we would leave the discussion section 4.4. as is. We will add the change lines 253–254 of our current methods section 2.3.8. to following to describe the adapted cross validation:

We used an adapted leave-one-out cross validation (i.e., leave-one-plot-out CV) to account for spatial autocollinearity within plots and thus to reduce overfitting (Yates et al. 2022). Iterating through all plots, only one plot and its observations were taken as

test data, while the rest of the dataset was used as a training set, until all plots were used once as test data, on which we tested the prediction accuracy of our model.

**Ref2C35:** A principal component analysis could also be informative to demonstrate the differences between your soil types / geochemical gradient and its relationship to your various bulk measurements, even if it's only in the supplementary information. This could also provide interesting insight into how your geochemical variables are associated with your C variables. Is this coming in a separate manuscript? If not, I'd suggest you include this and more figures in the supplementary information, exploring depth gradients and the distribution of your measured variables.

**Our response:** Thank you for this comment. We created a PCA biplot, which we will add as Figure S1 to our supplement. The biplot includes all measured bulk soil variables, including 'depth', a variable that corresponds to the lower depth of a respective horizon, measured from the soil surface, and SOC so that it becomes easier to see how geochemical variables are associated with depth and SOC.

The PCA including all measured bulk soil variables highlights the relationship of soil texture, elements, pedogenic oxides, and SOC with depth. As expected, SOC and depth point in opposite directions, indicating the negative relationship between SOC concentrations and depth. Furthermore, the colors of the parent materials show how each site is lined up along depth, from more shallow in the top left to deeper soil samples in the bottom right of the plot. Poorly crystalline Fe and Al ($Fe_{AO}$ and $Al_{AO}$) appear to be positively correlated with depth, given their proximity to depth within the PCA biplot. However, there seems to be no clear relationship of depth with organically-complexed Fe and Al ($Fe_{PP}$ and $Al_{PP}$).

[Figure]

**Fig S1.** Principal component analysis biplot displaying the first two principal components (PC1 and PC2) for measured bulk soil variables, including depth, SOC, N, Zr, P, Na, K, Ca, TRB, Si, Mg, Fe, $Fe_{DCB}$, $Fe_{ex}$, Al, $Al_{DCB}$, $Al_{ex}$, Mn, $Mn_{ex}$, $CEC_{eff}$, sand, and silt. Observations are shown considering individual parent materials, corresponding to different respective colours. Scaled loadings of individual variables are shown as arrows. The colour of these arrows corresponds to their relative contribution (%) to the respective PCs. Dolomite observations are excluded as its observations were not considered in the SOC model predictions. Please note that the depicted variable contributions were computed to quantify the relative importance of each variable in defining the two first PCs. These contributions were calculated as the sum of squared loadings for each variable across PC1 and PC2

We will add the following to the method section 2.3.8 to describe how the PCA biplot was computed:

We conducted a principal component analysis (PCA) to explore patterns and relationships among all measured bulk variables, SOC, and 'Depth', a variable capturing the lower depth of each horizon measured from the soil surface. All input variables were standardized using z-score normalization prior to analysis. The PCA was performed using the prcomp function of the R package *stats* (R Core Team 2023), which applies singular value decomposition (SVD) to extract principal components (PCs) that capture the major axes of variation in the data. The resulting eigenvectors (loadings) describe the correlation between each variable and a given PC. We also calculated variable contributions to PC1 and PC2 as the sum of squared loadings across these components, expressed as percentages. These variable contributions

served as visual indicators of each variable's relative importance in defining the PCA structure in the resulting biplot.

We will add a paragraph into our results subsection 3.1, where we summarize some of the most important findings from the PCA analysis. In this subsection we also suggest removing the subsubsection 3.1.1 soil fertility and 3.1.2 soil mineralogy to just have one joined subsection '3.1 Characterization of the soil profiles based on soil fertility and soil mineralogical properties'. We also suggested, in response to a later comment (see **Ref2C38**), that the lines 274–279 and 286–291 be removed and placed into the discussion subsection 4.1. We suggest adding the following paragraph into the 'new' subsection 3.1. following the first paragraph describing soil fertility trends (current lines 261–272) and soil mineralogy trends (current lines 281–285):

A PCA conducted on measured bulk soil biogeochemical variables, including SOC and depth (Fig. S1), revealed clear clustering by parent material, along a pH–C:N gradient. The C:N ratio aligns positively with exchangeable and pyrophosphate-extractable Fe and Al, whereas pH associates positively with DCB-extractable (i.e. more crystalline) Fe and Al phases. Along this gradient, Gneiss-derived soils form a clearly discernible group from the other parent materials. SOC is strongly associated with total N, sand content, and bioavailable P, while being negatively associated with depth, as expected. Contrarily, depth aligns positively with ammonium oxalate-extractable Fe and Al. Depth appears orthogonally to the pH–C:N gradient, suggesting that vertical depth-related changes in soils are structured independently of the major pH–C:N gradient. Lastly, depth structures observations within each site, as indicated with the coloration of parent material.

**Ref2C36:** SOC prediction models - what was the effect of the exclusion of clay content from your prediction model? It seems strange to include it in both the soil mineralogy and soil fertility models if it has such an a strong explanatory power. It seems more relevant to the soil mineralogy prediction model. Please test this or provide a strong rationale for its inclusion in both models.

**Our response:** Thank you for raising this question. Reviewer 1 raised a similar concern for the inclusion of clay in both models (see **Ref1C2**). However, to first address the question on what happens when clay is excluded, we reran the fertility model (leave-one-plot-out CV) without clay content below (see **Revised Figure 5. no clay variation (a))**. The fit decreased slightly with a previous $R^2$ of 0.86 down to 0.85 and a previous *RMSE* of 2.14 up to 2.19, compared to when it is included (**Revised Fig. 5.**, adjusted in response to Ref1C1). The relative variable importance previously explained by clay content was 'redistributed', mainly increasing the importance of the same top predictor, bioavailable phosphorus, from a previous 32% variable importance up to ~40% and a couple percent are allocated to all other variables.

[Figure]

**Revised Figure 5. (a)** Predicted bulk SOC (%) values vs. measured/observed bulk SOC (%) values of the soil fertility model. **(b)** Predicted bulk SOC (%) values vs. measured/observed bulk SOC (%) values of the soil mineralogy model. **(c)** Relative importance (%) of the model predictor variables included in the soil mineralogy model. **(d)** Relative importance (%) of the model predictor variables included in the soil fertility model. The dashed line represents a 1:1 regression line.

[Figure]

**Revised Figure 5. no clay variation (a)** Predicted bulk SOC (%) values vs. measured/observed bulk SOC (%) values of the soil fertility model, when clay is excluded as a predictor variable.

In response to the concern about including clay in both models as a predictor, we wanted to reiterate that the main incentive of our decision to design two separate prediction models for SOC with distinct sets of variables is to investigate whether SOC is more related to stabilization mechanisms (mineralogical variables) or to fertility parameters that may govern C input (fertility variables). Clay can be seen as both a proxy variable for C stabilization potential (Yu et al. 2022) and also as a proxy for the retention of water and nutrients in the soil, thus contributing to overall soil fertility (Kleber et al., 2015; Yu et al. 2022). Since clay content is a proxy for both and multifunctional in that sense, we decided to leave it in both models. Another problem of merging the entire set of predictor variables into one prediction model would likely result in overfitting the model prediction due to the very low degrees of freedom of the model (not enough observations). Although we already allude to clay as a proxy for the retention of nutrients (lines 240–242) in the methods of our manuscript, we do not explicitly address the reasoning behind the inclusion of clay in both of our models. We will therefore add the following statement after line 246:

"Clay is included in both models to act as a proxy for nutrient retention and water holding capacity of the soil (Yu et al. 2022), in the case of the soil fertility model, and as a proxy for SOC stabilization potential in the soil mineralogy model, as has been commonly done in SOC prediction models (Abramoff et al. 2021, Georgiou et al. 2022)." The two sets of predictor variables merged together into one prediction model would also likely cause overfitting issues for the predictions due to a resulting very low remaining degrees of freedom (not enough observations).

**Ref2C37:** Are your averages presented as ± standard error. It's not stated explicitly, except in captions from what I could see. Maybe I missed it, but then the point is it could be made clearer.

**Our response:** Thank you for this question. In line 224 we write the following: Data are presented as means of related replicates with standard errors, unless specified otherwise.' We will exchange the word 'mean' here for 'average'. Moreover, we will explicitly add how the averages are presented at the bottom of each figure and table caption so that it becomes more clear.

**Results:**

**Ref2C38:** Be careful with reporting discussion in your (separated) results section. As you have separated the sections, please keep references and interpretation in the discussion. See line 274:279 for an example.

**Our response:** Thank you for this comment. We will slightly modify lines 274–279 and place them into the discussion section 4.1 before the pre-existing first paragraph that is already there. We have also carefully reviewed the rest of the manuscript and found a similar issue for lines 286–291, which we will also place into the discussion section 4.1, following the inserted lines 274–279. Lastly, we would like to remove lines 306–309 from section 3.2 and discuss the influence of the SOC stock calculation method in a paragraph beginning after the current line 405 in section 4.2, which is also in the interest of reviewer 3. The text that we suggest adding can be found as a response to **Ref2C52 and Ref3C3.**

**Ref2C39:** Line 261: Are widest and narrowest the correct terminology here? I would suggest not at is refers to a range and it's counter to your observations and their variability, reported just after, as the Marl site has a higher variability. I believe it should be a higher or lower C:N ratio.

**Our response:** Thank you for this comment. We will change the terminology as you suggested to 'higher' and 'lower' C:N ratio, everywhere where needed.

**Ref2C40:** Line 269: Superscript -1.

**Our response:** Thank you for this comment. We will adjust this.

**Ref2C41:** Line 274: this is discussion as per my above major point and should be removed from the results section. Here I believe you are referring to better conditions for plant growth or something specific. What are ideal conditions? They are subjective.

**Our response:** Thank you for this comment. We will place this statement into the discussion section 4.1. We agree that the formulation 'ideal conditions' is in fact not ideal and subjective as you say. We would change this to say better/worse plant growth conditions'.

**Ref2C42:** Line 312: as the oldest values, do you believe there was any geogenic influence on the F14C values of the Cw horizon at your Marl site or are you certain that decarbonation was complete? Also your Oh horizon in the dolomite soil has an older F14C value than Ah horizons from other soil types? Could you please discuss this.

**Our response:** Thank you for this comment. As a pre-treatment to the AMS assessments of $F^{14}C$, all soil samples are acidified according to the methods described in Ramnarine et al. (2011), as was described in more detail above in response to **R2C26**, and which will be amended to the revised manuscript. However, we did not explicitly quantify the potential presence of geogenic organic carbon in any of our sites' soils. While small, we acknowledge that there may be a potential contribution of geogenic organic carbon in any soils formed on sedimentary rocks and that the likelihood of its contribution becomes greater the closer to the weathering front. Therefore, we cannot rule out the presence of geogenic organic carbon in the Flysch soil either. We will add a statement regarding the potential influence of geogenic C on the Marl site's apparent $F^{14}C$ to line 414 of our discussion:

'Furthermore, in the case of Marl, a sedimentary rock, a potential presence of geogenic organic carbon may be present that could make the in-situ SOC appear more persistent than if it were derived solely from biological inputs and pedogenic processes alone. Kalks et al., (2021) found that geogenic organic C influence is of particular importance in young soils on terrestrial sediments with comparatively low amounts of OC. Hence the impact of these potential inputs would be largest in the subsoil, closest to the weathering front. As geogenic organic carbon was not specifically quantified in our study, we cannot rule out its presence in any of the sampled soils formed on sedimentary rocks.'

Regarding the Oh horizon having an older $F^{14}C$ value than the other soils Ah and Bw horizons we will change and augment some lines in the 4.3 part of the discussion as follows:

The Marl and Gneiss sites hold the most persistent SOC of all sites in their subsoil horizons (Bw, Cw), when excluding the Dolomite site's single observation (Fig. 4 (a)). However, Dolomite's Oh horizon's $F^{14}C$ value appears to be smaller than the $F^{14}C$ values of all other site's Ah and Bw horizons, despite constituting an organic layer. We assume that the Oh horizon formed here does not decompose due to reduced turnover, caused by the cold temperatures and harsh environment of the alpine environment. The produced organic matter accumulates at this site because there are no subsoil layers it could be incorporated into, as these cannot form on the Dolomite. The reduced decomposition is also reflected in the overall thickness of the layer (12.9 ± 2.2 cm), which is nearly double as thick as all other sites' Ah horizons. Additionally, the calculated weathering proxy, $Fe_{DCB}/Fe_{tot}$, is highest at the Dolomite site, indicating that soils developed on dolomite are highly weathered (Figure 2).

**Newly added references:**

Kalks, F., Noren, G., Mueller, C.W., Helfrich, M., Rethenmeyer, J., Don, A.: Geogenic organic carbon in terrestrial sediments and its contribution to total soil carbon, SOIL, 7(2), 347-362, 10.5194/soil-7-347-2021, 2021.

**Ref2C43:** Line 319:324: I was surprised at how much of your bulk SOC was represented in the microaggregate fraction in these soil types. I guess it is inline with other observations from Chilean alpine grasslands using a similar fractionation scheme. It's also interesting that it remained relatively constant with depth relative to your two other fractions. This warrants more detailed discussion later in the manuscript.

**Our response:** Thank you for this comment. Indeed, the observations we report for our SA fraction C contribution to bulk SOC are within the range reported by Wasner et al. (2024). However, the variation of SA relative contribution to bulk SOC was larger in the soils they examined, than in our study, with values ranging from 15.3–97.0%. They also note that the relative contribution of SA to bulk SOC exceeded that of s+c in many soils of their gradient, corroborating our findings further. We discuss the similar findings of Wasner et al. (2024) in connection with their fractionation scheme and discuss the implications other fractionation schemes could have had on our results in response to **Ref2C60.** Furthermore, reviewer 1 asked us a similar question, regarding the large contribution of microaggregates and their contribution to bulk SOC. Here we provided additional text for the discussion section to examine the importance of Fe and (reactive) Fe pedogenic oxides for soil aggregation processes (see response to **Ref1C3).**

**Ref2C44:** Line 319: Is it an importance of a fraction to bulk SOC? Or the distribution of C amongst fractions (g g-1 or g 100 g-1).

**Our response:** Thank you for this comment. It is the importance of a fraction to bulk SOC, i.e. it quantifies how much the C contained within a fraction contributes to bulk SOC. To clarify this point, we will add the following to line 319:

The importance of a fraction's relative contribution to bulk SOC (%), i.e., how much the C contained within a fraction contributes to bulk SOC, varies within each site's soil horizons and across all sites (Fig. 4 (b)).

**Ref2C45:** Line 324: but increasing with depth.

**Our response:** Thank you for this comment, we will change the sentence to: "It therefore displays a trend opposite to that of the POM fraction but with increasing depth."

**Ref2C46:** Line 328: In your methods you refer specifically to prediction of SOC stocks, here you predict SOC content. Please be specific and consistent with your terminology throughout.

**Our response:** Thank you for this comment. We will change the terminology from 'SOC stocks' to 'SOC content' in the methods section 2.3.8, lines 235–327.

**Ref2C47:** Line 329: $R^2$ typically.

**Our response:** Thank you for this comment. We will change the notation and display it in an italicized manner.

**Ref2C48:** Line 332: Again, I'm not sure that you should include clay in both models, particularly if it's such an important predictor. I would suggest that you test its exclusion and then if you choose to include it, defend its inclusion in both models more thoroughly.

**Our response:** Thank you – we addressed this comment above in response to **Ref2C36**.

**Discussion:**

**Ref2C49:** As a general comment, it would be good to point the reader to any existing literature or data on these specific sites with a paragraph in the discussion, discussing the observations that have been made and how they compare with your data, to give the reader better context. It would also be informative to compare the results of authors who have used a similar fractionation scheme (building on your comments about Wasner et al., 2024) and compare the results of your fractionation, the distribution of SOC amongst fractions in your system with theirs, and discuss how differences in fractionation scheme or soil systems could have driven these observations.

**Our response:** Thank you for this valuable point. There are several studies with locations in the European alps, which we acknowledge in our introduction e.g. Hitz et al., 2001; Guidi et al., 2024; Canedoli et al., 2020; Cao et al., 2013) that however have different research foci. To our knowledge, the Flysch, Gneiss, Greenschist, and Marl sites in our study have not previously been sampled at the locations we visited and thus a comparison of our data with other data from these specific sites is unfortunately not possible. However, we tried to compare our plant biomass OC and SOC stock estimates to that of other studies conducted at similar altitudes and similar vegetation, in our discussion section lines 356–360.

To support our fractionation analysis (also in response to **Ref1C3**) and compare our results with Wasner et al., 2024 as well as other studies, we suggest to slightly change the phrasing discussion of the lines 442–446 and augment the discussion at this location of the manuscript with the discussion of additional literature as follows:

'Wasner et al. (2024) examined a geoclimatic gradient across a diverse range of grassland topsoils and found that the stable microaggregate fraction was the biggest contributor to bulk SOC in C-rich soils. They report that both the SOC quantity in free silt and clay (s+c) and stable microaggregates (MA) fractions were positively correlated to pedogenic oxide contents and texture. The results from this publication support the notion that stable microaggregates can be major contributors to bulk SOC in grassland soils across large environmental gradients. Another publication by Lehndorff et al., (2021) examined the spatial organization of soil microaggregates in a sandy and a loamy Luvisol. They report greater OC concentrations within the microaggregate fractions compared to bulk SOC, and their analyses support the notion

that OC forms the core for microaggregate formation and is protected within microaggregate structures. They also found a systemic increase in iron, followed by clay and silicate mineral phases in the microaggregates formed on clay-richer soil, indicating that the inherent soil mineralogy is reflected in the composition of microaggregates. Their data further supports the notion that pedogenic iron aids aggregation processes in the soil by acting as a cementing agent (Campo et al., 2024). The importance and prevalence of iron in microaggregates is therefore shown to be a supporting phase for overall microaggregate stability, and thus ultimately supports the retention of SOC therein. This high relevance of Fe for aggregation processes is also reflected in significant correlations of reactive Fe pedogenic oxides ($Fe_{PP+AO}$) and total Fe pedogenic oxides $Fe_{(PP+AO+DCB)}$ with the MA mass (wt%) for our sites (both have a correlation coefficient of 0.48 with $p < 0.05$; Fig S2). While no significant correlations could be found for reactive or total Al or Mn pedogenic oxides (data not shown). Due to an overall significantly lower amount of total Mn and extractable Mn pedogenic oxide concentrations, we assume the contribution thereof to be rather little to aggregation processes (data not shown). We interpret the insignificant connection of Al pedogenic oxides to MA mass to be related to the pH range of most of our sites, which is above values where Al becomes significantly mobilized and ranges generally between 4.2–6.8, except for the Granite site (3.8 ± 0.2 (SD)).

**Ref2C50:** Line 371:373: This is surprising, I wonder how much of it is related to the fact that shrubs are pioneer species that prefer sandier soils, and produce more biomass in your system, relative to a stabilising influence of sand on high OC stocks (black sand soils being a notable exception).

**Our response:** Thank you for this comment. The same study that we cite in our manuscript demonstrates that the changes in SOC in a soil following shrub encroachment are soil texture-dependent, climate and shrub species dependent (Li et al., 2016). However, we are not trying to imply that sand has a high stabilizing influence on the incoming SOC from encroached shrubs. On the one hand we aim to convey that this particularly sandy site demonstrates conditions ideal for the genus *Vaccinium* to encroach, with its acidic pH range and sandier texture ensuring better water drainage (Nestby et al., 2011; Ritchie 1956). On the other hand–as you are suggesting–we are trying to say that in the case of such sandy-textured soils, it has been shown that shrub encroachment can have a positive effect on SOC (Li et al., 2016). We will clarify this in the next manuscript version.

To convey this message more clearly, we would like to change the lines from the current 344 onwards to the following:

The soil conditions present at the Gneiss site, such as an acidic pH range and sandy texture, providing good water drainage, benefit the encroachment and growth of shrubs of the *Vaccinium* genus (Chen et al., 2019; Nestby et al., 2011; Ritchie 1956). The growth of these shrubs contributes to the very high percentage of woody biomass and large C:N ratios at this site (Fig. 3 (a), Table 4). Furthermore, shrub encroachment

in sandy-textured soils can have beneficial effects on SOC, given the higher biomass input (Li et al., 2016).

We would also like to rewrite the lines 371–373 to the following so that the message stated above comes through more clearly here again:

This shrub-induced increase in SOC stocks is presumably linked to the high sand content found at the Gneiss site, which provided ideal conditions such as an acidic pH and good water drainage, for the encroachment of shrubs from the *Vaccinium* genus. A meta-analysis of worldwide shrub-encroached grasslands showed that resulting SOC content changes were soil texture dependent, with decreases in silty and clay soils and increases in sandy soils (Li et al., 2016). However, the meta-analysis also shows that SOC content changes following encroachment were genera-dependent. Lastly, they found that the main drivers behind SOC content gain or losses with shrub encroachment were the relative differences in productivity between shrub-encroached vs. non-shrub-encroached grasslands.

The lines following the current line 346 would remain in their current state.

**Newly added references:**

Li, H., Shen, H., Chen, L., Liu, T., Hu, H., Zhao, X., Zhou, L., Zhang, P., and Fang, J.: Effects of shrub encroachment on soil organic carbon in global grasslands, Sci. Rep., 6, 28 974, https://doi.org/10.1038/srep28974, 2016.

Nestby, R., Perical, D., Martinussen, I., Rhloff, J.: The European blueberry (Vaccinium myrtillus L.) and the potential for cultivation. A review, EJPSB, 5, 5-16, 2011.

Ritchie, J.C.: Vaccinium Myrtilus L., Journal of Ecology, 44(1), 291-299, https://doi.org/10.2307/2257181, 1956.

**Ref2C51:** Line 375: This could be a good break of a paragraph rather than Thus, and Therefore, in subsequent sentences.

**Our response:** Thank you for this comment. We will adapt this into our manuscript and create a paragraph break. Then we will also remove the 'thus' and 'therefore' as suggested.

**Ref2C52:** Line 378: How much of this is driven by the soil thickness and stage of subsoil development, relative to a purely geochemical influence.

**Our response:** Thank you for this comment. The way SOC stocks are calculated are very much dependent on horizon-specific and whole-profile soil thickness. However, depending on the parent material, the respective soil horizons have developed different concentrations which also contribute to SOC stock calculation. Both the thickness of respective soil horizons and whole profiles as well as average concentrations of specific geochemical parameters are a function of subsoil development. In the manuscript we currently acknowledge the influence of soil thickness on SOC stock calculation in lines 305–309, however, we decided to remove

these lines from section 3.2 as mentioned in a response to Ref2C38, as we wanted to discuss the influence of the SOC stock calculation method in a paragraph beginning after the current line 405 in section 4.2, which is also in the interest of reviewer 3. We provide the following a paragraph summarizing SOC stock calculation methods as a response to **Ref3C3**:

In our study, as a result of our calculation methodology (horizon specific sampling and analyses instead of fixed depth increments), the magnitude of a horizon-specific and whole profile SOC stocks depend on the thickness of individual soil horizons and their respective OC contents (see Sect. 2.3.7). Thus, because thick horizons (such as Bw) can harbor more C as part of the profile stock even if C contractions are lower than in thinner (topsoil O or A) horizons (Fig. 2, Sect. 2.1). Thus, if SOC stocks were normalized to the same depth increment thickness, the trends reported here would potentially change. Furthermore, SOC stock estimates can vary significantly depending on the calculation method applied. For example, Poeplau et al. (2017) show that discrepancies in estimated SOC stocks between methods increase with increasing rock fragment content. In soils with rock fragments comprising > 30 vol %, they revealed that SOC stocks may be overestimated by as much as 100%. Thus, as our sampling took place in steep terrain with strongly varying rock fragment contributions of 1.6 ± 0.9 in the Dolomite's Oh, up to 52.5 ± 2.6 for Greenschist's Cw horizon (see Table 4.), we calculated SOC stocks by taking these varying contributions into account, as to not significantly overestimate SOC stocks by applying the calculation methodology as suggested by Poeplau et al. (2017) (Sect. 2.3.7).

**Ref2C53:** Line 381: Relatively speaking, dolomite is less soluble than calcite, another important mineral in alpine grasslands in the Alps.

**Our response:** Thank you for this comment. We will leave our original statement intact but add an acknowledgement that calcite represents an important mineral in alpine grasslands in the Alps and would be a mineral of even higher solubility.

**Ref2C54:** Line 389: Does the combination of PP and AO not contradict the findings in Line 335 - that the poorly crystalline AO oxides correlated negatively with SOC, while PP positively correlated with SOC prediction in your model?

**Our response:** Thank you for this comment. PP and AO individually behave differently than when they are summed together. Since the positive correlation of PP is stronger than the negative signal of AO (see Fig. B2), it does not seem to contradict the findings in Line 335.

**Ref2C55:** I would suggest including more figures, at least as supplementary figures to help display the distribution of your bulk geochemical data and its correlative relationship with SOC, it may help the reader understand the distribution of your data.

**Our response:** Thank you for this comment. We have one supplementary figure that displays the density distribution, correlation coefficients and scatterplots of various bulk soil parameters (those considered in our two statistical models) and SOC content.

We will be adding a supplementary figure of a PCA biplot that shows the relationship between all bulk soil parameters measured in our dataset (including some additional variables not included in the modelling) with soil depth and SOC. Please see our response to **Ref2C35** for more details regarding the PCA.

**Ref2C56:** Line 412: I hope that the authors have collected F14C data on these fractions as it would be great to see this in a subsequent manuscript.

**Our response:** Thank you for this comment. We currently have not collected F14C data on the examined fractions in this manuscript but plan to do so in future work as we intend to advance the development of soil C turnover models.

**Ref2C57:** I wonder again if this could be related to an incomplete carbonate removal in the Marl site.

**Our response:** Thank you for this comment. As we mentioned earlier, we have strong reasons to assume that our decarbonation protocol worked and that afterwards no more carbonates were present in our Marl samples.

**Ref2C58:** Line 422: I think that this is "rather stabilization-[driven] than input-driven, adding the hyphen creates a connection to -driven, otherwise it reads slightly bizarrely.

**Our response:** Thank you for this comment. We will add the hyphen as suggested.

**Ref2C59:** Line 439: I don't believe this agrees with your abstract? Please be consistent with the use of numbers and terms throughout.

**Our response:** Thank you for this comment. We will adapt the phrasing here and, in the abstract, to make sure that the same numbers and underlying message are presented. We will change the phrasing in the abstract to say the following:

'We show potential differences in the importance of SOC stabilization mechanisms, with the microaggregate soil fraction contributing ≥ 50% to bulk SOC in most cases.

In line 439 we would suggest changing the wording to the following: In general, the MA fraction represents the most important SOC stabilization fraction, contributing ≥ 50 % to bulk SOC (%) in the majority of observations, across all sites (Fig. 4, MA). With the exception of the Ah horizon at the Gneiss and Marl sites.'

**Ref2C60:** Line 446: It seems that this is a rather important part of your discussion that stops abruptly. I would suggest building on this and exploring its connection not only to Wasner et al., who used a similar fractionation scheme, but also other studies that have used different fractionation schemes to explore its relevance and reasons. Every fractionation scheme results in operationally defined pools, but it would be good to discuss the relevance of this and compare it to other studies and systems.

**Our response:** Thank you for this valuable feedback. We will augment and adapt this part of the discussion to the following:

'A large importance of MA to bulk SOC was also found in Chilean grassland topsoils under cold climates with high bulk SOC contents, in a study that applied a similar fractionation scheme, combining size and density fractionation steps (Wasner et al., 2024). SOC quantity in both s+c and MA fractions was also found to be positively correlated to pedogenic oxide contents and texture in their inspected soils. Though we did not conduct any analyses that would inform us on the spatial organization of SOC and elements surrounding and comprising MA and s+c, this would have potentially provided valuable, mechanistic insights as to how SOC is stabilized within these fractions in the context of the examined alpine soils.

Leuthold et al. (2024) compared several common fractionation methodologies including 1) simple size separation, 2) density flotation and 3) two different combined size and density on a variety of agricultural soils across the continental United States to identify chemical similarities and differences across the methodologically defined and -derived fractions. These include mineral-associated organic matter (MAOM), particulate organic matter, (POM) and coarse heavy associated organic matter (CHAOM). Whereby, their MAOM fraction most closely represents the s+c fraction and their CHAOM fraction the MA fraction extracted in our study. Their results showed that across fractionation schemes, MAOM fractions were consistent in terms of their spectral, isotopic and chemical characteristics, but that the POM fraction was highly variable. The resulting C concentrations in POM ranged between $1.3 \pm 0.2$ to $3.9 \pm 0.7$ mg C g soil$^{-1}$ across the second and first combined size and density fractionation methods, and were more comparable between the one-step methods. MAOM C concentrations were similar across fractionation methods but were slightly higher in one-step methods. The fractionation method applied in this study was similar in nature to the first of the combined size and density fractionation methods examined by Leuthold et al. (2024). However, the MA and s+c fractions in our study were defined solely by size, following the physical breaking up of macroaggregates, rather than by size and density. Considering the results of Leuthold et al. (2024), our fraction-specific SOC results should be seen as a function of the applied fractionation method. Accordingly, our estimated POM C concentrations may have resulted in larger values than if a different kind of fractionation protocol. Further, depending on the applied protocol, no CHAOM or MA fraction is isolated. This also leads to a different importance attributed to the POM vs. MAOM relative C contributions to bulk SOC. Leuthold et al. (2024) stress that further research into the CHAOM fraction is needed, whose stoichiometry and isotopic signature distinguish it somewhat from the MAOM.'

**Figures & Tables:**

**Ref2C61:** General: Some tables are included in the text and others not, I suspect this is just a formatting issue in early submission. Table 4 pops up earlier in the text then the other tables, and the table numbers should instead increase sequentially. Unless this data is being explored in a subsequent manuscript or is already under review, please consider adding more supplementary figures to explore your bulk geochemistry dataset.

**Our response:** Thank you for this comment. The positioning of some of the tables is indeed a formatting issue of this preprint manuscript version. We will augment the supplementary with a figure displaying the changes of selected bulk geochemical variables and a PCA biplot.

**Ref2C62:** Fig. 1: I would suggest dropping the colour in the map background to more clearly highlight where the samples come from. A grey scale for the coarse hillshade/digital elevation model in the background would highlight the site colours more clearly.

**Our response:** Thank you for this comment. We have provided a version of the map in a grey scale to better accommodate this and will replace the current Figure 1. with this revised version.

[Figure]

**Revised Figure 1.** Locations of the soil sampling sites developed on distinct parent materials. This map was produced using Copernicus WorldDEM-30 © DLR e.V. 2010-2014 and © Airbus Defence and Space GmbH 2014-2018 provided under COPERNICUS by the European Union and ESA; all rights reserved (ESA, 2023).

**Ref2C63:** Fig. 2: As a suggestion, rather than reporting the rooting zone, do you think it would be interesting to present root coverage as a relative percentage of your profile surface with depth? Don't let this suggestion crowd your figure unnecessarily, but it could be informative.

**Our response:** Thank you for this comment and interesting idea. We would like to keep the main message of the figure rather clear; we would like to focus on the horizonation and soil development depths of the different soil profiles. We do not have finely enough resolved root data to accurately capture the root coverage as the relative percentage of our profile surface. However, we did measure overall rooting depths and we included it as a simple dashed line so that readers could get a rough insight into how far roots manage to intrude into the soil profiles. Therefore, we would like to keep the Figure the same as it currently is.

**Ref2C64:** Table 1: I'd add slope aspect and character to this table. Even if they were controlled for, it would be nice to present them and how this could have influenced your interpretations.

**Our response:** Yes, we will add slope and aspect to Table 1. We do not understand what was meant with 'character' so we sadly cannot comment on this.

**Ref2C65:** Fig. B1 & B2, can you please remove the grid lines from the plots in the upper right so it's easier to read the parent material type and numbers.

**Our response:** Yes, thank you for pointing this out. We will remove the grid lines to enable better visibility.

---

## Author Comment (AC3)

**Author responses to comments for Maier et al. Biogeosciences preprint manuscript**

**Reviewer 3, Frank Hagedorn:**

**Overview Ref 3:** The manuscript by Maier et al. represents a comprehensive assessment on SOC storage in alpine grassland soils, clearly demonstrating that parent material geochemistry plays a dominant role in controlling SOC stocks; an aspect often overlooked in the current literature.

**Our response:** We appreciate Frank Hagendorn's support for our manuscript and his constructive feedback on an important topic that helped improve the discussion. We have provided a detailed point-by-point response below and highlight, text that we will add to the revised manuscript version in red.

**Ref3C1:** Although alpine soils are known to be particularly stony, it remains unreported if and how coarse stone fragments have been incorporated in the estimates of SOC stocks. The authors state that soil sampling was conducted using Kopecky cylinders (100 cm³) within soil profiles and Edelman augers to bedrock depth. Although this method permits estimation of bulk or fine earth density within the cylinder volume, it does not capture larger stone fragments present throughout the profile. In stony soils, typical for alpine environments, omitting these coarse fragments can lead to substantial overestimation of SOC stocks (Poeplau et al., 2017). Given the frequent absence of direct measurements of coarse fragments, many studies apply corrections to SOC stock estimates based on field-derived stone content data. It is therefore recommended that the authors clarify whether such corrections were applied, and if not, to discuss the implications for their SOC stock estimates.

**Our response:** Thank you very much for your valuable comments. We did account for the coarse fragments in the estimates of SOC stocks, by using the suggested SOC stock calculation from Poeplau et al. (2017). However we acknowledge that our description of the SOC stock calculations should be more clear. We would therefore like to adapt our methodology description in section 2.3.7. In our manuscript the stocks were corrected for the stone content as follows: We subtracted the coarse fragment fraction from the soil mass, yet used the 'full' soil volume, to compute the SOC stocks. Our estimates therefore take the coarse fraction mass and volume into account as to attempt to not overestimate SOC stocks (lines 217-220 and equation (2)). We will however discuss the SOC stock estimates. For further transparency, we will provide the average ± SE of coarse fragment content per horizon and site in Table 4 of the revised manuscript. Upon double checking we also realized that the equations we are quoting from Poeplau et al. (2017), were in fact not their eq. 3 but their equations 7 and 8, thus we will correct this in the revised version. We will define the terms used in equation (2) and adapt the description of eq. 2 and 3. as follows from line 218 onwards to, clarify how we accounted for the coarse fragments:

'To account for coarse fraction contained in our soil samples in our SOC stock estimations, respective SOC horizon stocks ($SOC_{stockj}$) were calculated in two steps following Eq. 7 and 8 proposed by Poeplau et al. (2017). In a first step, what they call the fine soil stock for a respective horizon (j) is estimated ($FSS_j$), (Eq. 2), where the $mass_{finesoil}$ is the mass of the total sample without the mass of the coarse fraction, $volume_{sample}$ is the total volume of the sample including the coarse fragments and $depth_j$ is the thickness of a respective horizon. In a second step SOC stocks are calculated for a respective horizon ($SOC_{stockj}$) (Eq. 3), where the $SOC_{concfinesoil}$ is the OC concentration of the fine soil multiplied with the previously calculated fine soil stock $FSS_j$ (i.e. the total sample without the mass of the coarse fraction). Whole profile SOC stocks were calculated by summing all horizon specific SOC stocks per sampling site.

**Ref3C2:** Additionally, reporting the slope of each soil profile is important for enabling surface-area-based corrections of SOC stocks (Prietzel & Wiesmeier, 2019).

**Our response:** Thank you for this comment. We will add the slopes of each soil profile to Table 1.

**Ref3C3:** Accurate sampling methods and SOC stock estimations are essential for enabling meaningful comparisons across ecosystems and for supporting upscaling efforts. A critical discussion of the uncertainties associated with SOC stock estimates would be valuable, even though these uncertainties do not undermine the manuscript's central finding that parent material geochemistry is a key control on SOC storage.

**Our response:** Thank you for this comment. We will provide an additional segment in the discussion on the uncertainties associated with SOC stock estimates based on the coarse rock fraction, the potential influence of geogenic organic carbon (see response to reviewer 2's comment on Line 342, **Ref2C42**) and the general difficulty to identify representative and relatively undisturbed plots in high alpine environments.

We first of all suggest to remove the text currently placed at lines 306–309, from the sentence beginning with "These findings are linked to the SOC stock calculation methodology, […]". We would however like to add the sentiment of these lines into our discussion, before line 405 in the following way:

In our study, as a result of our calculation methodology (horizon specific sampling and analyses instead of fixed depth increments), the magnitude of a horizon-specific and whole profile SOC stocks depend on the thickness of individual soil horizons and their respective OC contents (see Sect. 2.3.7). Thus, thick horizons (such as Bw) can harbor more C as part of the profile stock even if C concentrations are lower than in thinner (topsoil O or A) horizons (Fig. 2, Sect. 2.1). If SOC stocks were normalized to the same depth increment thickness, the trends reported here would potentially change. Furthermore, SOC stock estimates can vary significantly depending on the calculation method applied. For example, Poeplau et al. (2017) show that

discrepancies in estimated SOC stocks between methods increase with increasing rock fragment content. In soils with rock fragments comprising > 30 vol %, they revealed that SOC stocks may be overestimated by as much as 100%. Since our sampling took place in steep terrain with strongly varying rock fragment contributions of 1.6 ± 0.9 in the Dolomite's Oh, up to 52.5 ± 2.6 for Greenschist's Cw horizon (see Table 4.), we calculated SOC stocks by taking these varying contributions into account by applying the calculation methodology as suggested by Poeplau et al. (2017) (Sect. 2.3.7), as to not overestimate SOC stocks. In addition to uncertainties related to the SOC stock calculation methodology, spatial heterogeneity of SOC further introduces uncertainty. Due to the formation of microenvironments, soil-forming processes can vary at very small scales (Körner, 2003; Kemppinen et al., 2024). We aimed to minimize this effect by collecting composite samples which consisted of 10 individual samples per horizon and plot (Method section 2.2). Even with these measures–adjusted SOC stock calculation method and composite sampling–inherent uncertainties associated with spatial heterogeneity of SOC will remain part of studies that estimate SOC stocks at a regional scale.

**Newly added references:**

Kempinnen, J., Lembrechts, J.J., Van Meerbeek, K., Carnicer J., et al.: Microclimate, an important part of ecology and biogeography, Glob. Ecol. Biogeogr., 33(6), 313834, 10.1111/geb.13834, 2024.

---

## Referee Report (RR1)

**Referee 2:**

I thank Maier et al. for their careful consideration of the reviewers' comments and for the revisions they have made. I have only a few minor comments remaining, and once these are addressed, the manuscript will be ready for publication in *Biogeosciences*. Thank you again to the authors.

**Ref2C18:** Line 94: Sorry if I wasn't clear in my communication. Calcaric Cambisol is correct terminology. Calcic Cambisol was not. You had named your profile a Calcic Cambisol, I was suggesting to change it to: **Calcaric Cambisol**.

Ref2C64: "and character" was a typo. Sorry for the confusion.

**New minor comments:**

Author names: there is an extra comma in: Maria, E.

Line 26: Plant residues

Line 53: is it: studying elevational SOC gradients and? instead of studying studying.

Line 160: Aliquots.

Line 202: After, 5–10 min of cooling outside the bath, the vials were centrifuged

Line 234:235: 2 x pedogenetially? As in pedogenetically?

Line 355: Shwon

---

## Author Response (AR2)

**Author responses (AC) to comments for Maier et al. manuscript 21.10.2025**

Dear editors,

Thank you again for giving us the opportunity to address the minor comments on the preprint draft of the manuscript "Parent material geochemistry – and not plant biomass – as the key factor shaping soil organic carbon stocks in European alpine grasslands" for publication in the Journal of Biogeosciences. We appreciate the additional time and effort that you and the reviewers dedicated to providing a second round of feedback on our manuscript and are grateful for the attentive reading that went into identifying typos and minor errors. We have addressed all of these typos and minor comments. All changes are highlighted in a track-change file. Please see below, an additional response to reviewer 2's comments.

**Referee 2:**

**Overview Ref 2:** I thank Maier et al. for their careful consideration of the reviewers' comments and for the revisions they have made. I have only a few minor comments remaining, and once these are addressed, the manuscript will be ready for publication in Biogeosciences. Thank you again to the authors.

**Our response:** We appreciate the positive assessment and support from Reviewer #2 on our revised manuscript and thank them for their identification of typos and their addition of select comments. We have implemented all of their comments and suggestions into the second version of our revised manuscript. Furthermore, we again carefully reread the entire manuscript and identified further small errors in the form of typos, individual incorrect Figure/Table references and also changed the model performance metrics contained in Table B1 for the leave-one-out cross validation random forest model. The prior version of the manuscript contained the values from the 'original' version of the random forest model we had applied, which did not account for spatial autocorrelation as well.

---

## Author Response (AR3)

**Author responses (AC) to comments for Maier et al. manuscript 03.11.2025**

Dear editors,

Thank you for accepting the manuscript "Parent material geochemistry – and not plant biomass – as the key factor shaping soil organic carbon stocks in European alpine grasslands" for publication in the Journal of Biogeosciences. We have found 1-2 additional typos and changed them. Thank you for your time and effort in the processing, reviewing, editing and publishing of this manuscript.